# Single-cell transcriptomics of the goldfish retina reveals genetic divergence in the asymmetrically evolved subgenomes after allotetraploidization

Tetsuo Kon[1,9], Kentaro Fukuta[2], Zelin Chen[3,10], Koto Kon-Nanjo[4], Kota Suzuki[5], Masakazu Ishikawa[6], Hikari Tanaka [6], Shawn M. Burgess [3], Hideki Noguchi[2,7], Atsushi Toyoda [7,8] & Yoshihiro Omori [1✉]

The recent whole-genome duplication (WGD) in goldfish (*Carassius auratus*) approximately 14 million years ago makes it a valuable model for studying gene evolution during the early stages after WGD. We analyzed the transcriptome of the goldfish retina at the level of single-cell (scRNA-seq) and open chromatin regions (scATAC-seq). We identified a group of genes that have undergone dosage selection, accounting for 5% of the total 11,444 ohnolog pairs. We also identified 306 putative sub/neo-functionalized ohnolog pairs that are likely to be under cell-type-specific genetic variation at single-cell resolution. Diversification in the expression patterns of several ohnolog pairs was observed in the retinal cell subpopulations. The single-cell level transcriptome analysis in this study uncovered the early stages of evolution in retinal cell of goldfish after WGD. Our results provide clues for understanding the relationship between the early stages of gene evolution after WGD and the evolution of diverse vertebrate retinal functions.

[1] Laboratory of Functional Genomics, Graduate School of Bioscience, Nagahama Institute of Bioscience and Technology, Nagahama, Japan. [2] Center for Genome Informatics, Joint Support-Center for Data Science Research, Research Organization of Information and Systems, Mishima, Japan. [3] Translational and Functional Genomics Branch, National Human Genome Research Institute, Bethesda, MD, USA. [4] Department of Neurosciences and Developmental Biology, University of Vienna, Vienna, Austria. [5] Yatomi Station, Aichi Fisheries Research Institute, Yatomi, Japan. [6] KOTAI Biotechnologies, Inc., Suita, Japan. [7] Advanced Genomics Center, National Institute of Genetics, Mishima, Japan. [8] Comparative Genomics Laboratory, National Institute of Genetics, Mishima, Japan. [9] Present address: Department of Neurosciences and Developmental Biology, University of Vienna, Vienna, Austria. [10] Present address: CAS Key Laboratory of Tropical Marine Bio-Resources and Ecology, South China Sea Institute of Oceanology, Chinese Academy of Sciences, Guangzhou, China. ✉email: omori4@nagahama-i-bio.ac.jp

Whole-genome duplication (WGD) refers to the doubling of the entire genome during evolution that can cause massive diversification and evolutionary novelties[1]. Duplicated genes in a WGD (ohnologs) undergo various types of evolutionary fates after WGD[2]. The most frequent outcome is nonfunctionalization of one ohnolog member due to lack of selective constraint on preserving both ohnologs. In this case, ohnolog copies show biased expression levels. Other fates include sub-functionalization which preserve duplicates by partitioning the ancestral gene functions between copies, and neo-functionalization which assign novel function to an ohnolog copy. Ohnologs in both sub-functionalization and neo-functionalization fates display diversified expression among various cell types. In addition, dosage selection results in preservation of duplicates to maintain a dosage balance between interconnected components, leading to even expression profiles among ohnologs.

Vertebrates such as teleost fish, are thought to have experienced two rounds of WGD (1 R and 2 R) ~530 and 560 million years ago (MYA), respectively[3–6]. A teleost-specific third-round WGD (Ts3R) occurred in a common ancestor of teleost fish ~320–350 MYA[7,8]. As old events, many of the duplicated genes have been lost in the current vertebrate genomes, making it difficult to determine the early fates of ohnologs after WGD. Interestingly, a common ancestor of goldfish and common carp underwent a fourth round of WGD (Cs4R, carp-specific WGD) ~8–14 MYA[9–13], which makes it a useful model for studying the early fates of duplicated genes after WGD using genome and expression profiling studies.

The WGD in goldfish ancestor is considered to be an allotetraploidization event, which is the doubling of a complete set of chromosomes following the interspecific hybridization of diploid progenitors[14]. Asymmetric evolution of these duplicated subgenomes have been reported in certain organisms after the WGD events[15,16]. One of the duplicated subgenomes is often preserved in the ancestral state as a dominant subgenome, whereas the other nondominant subgenome experiences more chromosomal rearrangement, gene loss, and diversified gene expression[17,18]. Asymmetric subgenome evolution has been observed in domesticated plants[15,16,19,20], African clawed frog (*Xenopus laevis*)[18], and common carp (*Cyprinus carpio*)[21,22].

Goldfish harbors L-subgenome and S-subgenome, which are two asymmetrically evolved subgenomes. Gene selection and variation are observed more frequently in the nondominant S-subgenome than in the L-subgenome[23–25]. Previously, we reported a greater number of mutations linked to morphological phenotypes of goldfish strains in the S-subgenome[23]. The frequency of gene expression variation in a copy of duplicated genes is thought to be even higher, with a previous study reporting ~30% of duplicated genes with diversified expression profiles in seven tissues[11]. However, normal expression profiling analysis using bulk tissues (bulk RNA-seq) only identified the average number of transcriptional signatures across all cell types in tissues, suggesting the loss of information on cellular heterogeneity in tissues. As a result, the evolutionary diversification of ohnolog expression patterns at single-cell resolution in highly specialized cell types remain unclear. Recent advances in RNA sequencing and microfluidic platforms have dramatically enhanced the accessibility of single-cell transcriptomics with increased throughput[26].

The vertebrate retina is a light-sensitive tissue consisting of seven major classes of neurons (rod/cone photoreceptor cells, horizontal cells, bipolar cells, ganglion cells, and GABAergic/glycinergic amacrine cells) and three classes of glia (Müller glia, microglia, and oligodendrocytes) accompanied by retinal pigmented epithelial cells and vascular endothelial cells[27]. The evolutionary diversity in visual functions, such as wavelength shift, photoreceptor distribution, and retinal layer structure, is widely observed in vertebrates[28–31]. The highly structured retinas found in the current vertebrates are not found in the current invertebrate chordates that did not experience the two rounds of WGD[32]. Therefore, WGD is most likely to have contributed to the evolution of retinal diversification in vertebrates, however, a detailed analysis on the process of gene evolution during the early stages after WGD remains unclear[33,34]. In this study, we focused on single-cell transcriptome analysis of the retina for the following three reasons. First, in the previous study, the retina and the brain showed the most diversified expression between ohnologs after the recent WGD[23]. The retina has a wide variety of cell types expressing cell type specific genes with simpler tissue structure[35]. Second, to date, single-cell RNA-seq (scRNA-seq) analysis has been used to study the retina of various vertebrate species, including zebrafish, chicken, mice, and primates[26,36–41]. These studies provide datasets for the cell-type specificity of thousands of genes and demonstrate that the retina is an ideal tissue for studying the cell specificity of gene expression in vertebrates. Third, vertebrate retinas have highly diversified functions despite maintaining common basic cellular components and tissue structure, better allowing to compare the expression patterns between species.

In this study, analyses of gene expression and open chromatin regions were conducted at single-cell resolution. Unlike previously, we identified changes in gene expression between the L- and S-subgenomes after WGD with high accuracy. Notably, the gene expression changes between ohnologs on a larger scale than previously thought.

## Results

**Defining ohnolog pairs on the L- and S-subgenomes**. The goldfish genome consists of 50 chromosomes, which can be divided into 25 chromosomes (L-chromosomes and S-chromosomes), with each group corresponding to the chromosomes of the two progenitor species involved in Cs4R, an allotetraploid event[11,23] (Supplementary Fig. 1a, left panel). We termed the ohnologs located on the L-chromosomes the L-ohnologs and those located on the S-chromosomes the S-ohnologs. The chromosome set consisting of the 25 L-chromosomes is called the L-subgenome, and that consisting of the 25 S-chromosomes is called the S-subgenome[23]. We newly defined high-quality ohnolog pairs in goldfish to analyze preservation and divergence of L-ohnolog and S-ohnolog gene expressions using the scRNA-seq dataset (Supplementary Fig. 1a, right panel). We first identified the ohnolog candidates of 23,438 genes on the L-chromosomes and 20,666 genes on the S-chromosomes. Subsequently, we conducted a reciprocal BLAST analysis between these two goldfish gene groups and all zebrafish genes (23,651 genes). Consequently, we identified 11,444 ohnolog pairs (22,888 goldfish genes) with a high degree of confidence (Supplementary Fig. 1b–f; Supplementary Data 1). Thus, these 11,444 ohnolog pairs in goldfish have 11,444 corresponding zebrafish orthologs, located on their orthologous chromosomes (Supplementary Fig. 1c–f), which allow the inference of the biological functions of goldfish ohnologs by referring to previous studies using zebrafish. The 11,444 ohnolog pairs identified were improved compared with the 5404 ohnolog pairs that were analyzed in our previous study[23].

**Generation of the single-cell transcriptomic atlas of the goldfish retina**. To generate the single-cell retinal transcriptomic atlas, we used the retina from Wakin, a common goldfish strain, that retained wild goldfish features except for its body coloration[13]. We prepared single-cell suspensions from the dissected retina via

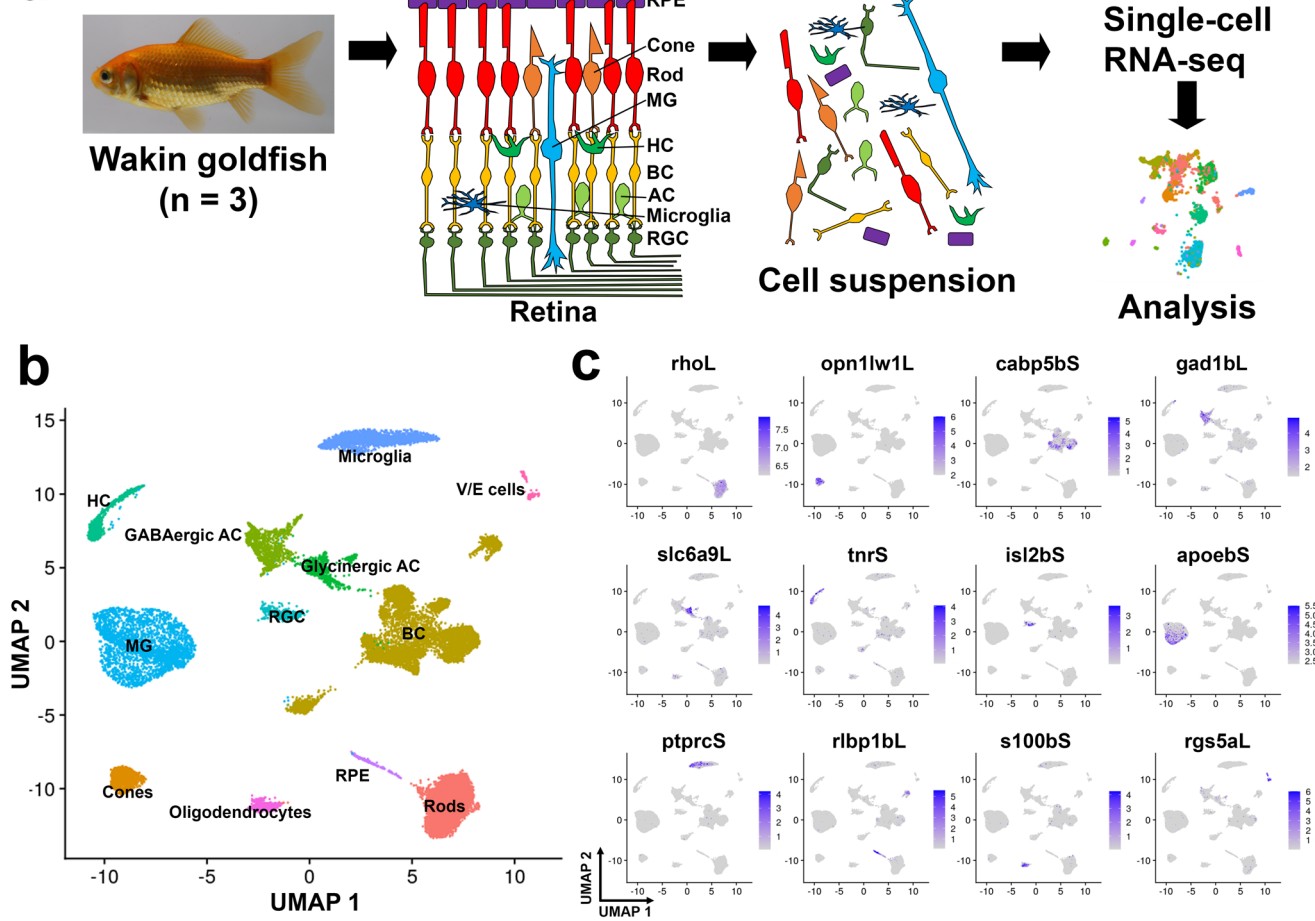

**Fig. 1 scRNA-seq analysis of the Wakin goldfish retina. a** The retinas from three Wakin goldfish individuals were enzymatically dissociated, and single cells were isolated, followed by scRNA-seq library preparation. The photo was taken by the authors. All figures were produced by the authors. **b** UMAP plot showing the cellular composition of the retina from Wakin goldfish ($n = 3$ biologically independent samples). The 22,725 cells were projected into a two-dimensional space by UMAP. The cells were classified into 12 cell types. **c** The expression of the cell-type-specific marker genes in the 12 goldfish retinal cell types is shown. RPE retinal pigmented epithelial cells, RGC retinal ganglion cells, HC horizontal cells, AC amacrine cells, BC bipolar cells, MG Müller glia, V/E cells vascular endothelial cells.

enzymatic digestion. We prepared scRNA-seq libraries using the 10× Genomics Chromium system, and sequenced them (Fig. 1a). After the data preprocessing and quality control, 22,725 cells were retained for downstream analysis ($n = 3$). We projected this data set into the two-dimensional space using the uniform manifold approximation and projection (UMAP) method. For this analysis, we used the publicly available scRNA-seq data of the zebrafish retina as a reference[42]. Therefore, we identified 12 cell clusters from the goldfish retina (Fig. 1b). Based on the gene expression of the known cell-type markers of the vertebrate retina[36–38,42,43], we identified 12 types of retinal cells, including seven types of neurons (rod photoreceptor cells [5842 cells, 25.7%], cone photoreceptor cells [911 cells, 4.0%], bipolar cells [7080 cells, 31.2%], GABAergic amacrine cells [1064 cells, 4.7%], glycinergic amacrine cells [895 cells, 3.9%], horizontal cells [668 cells, 2.9%], and retinal ganglion cells [338 cells, 1.5%]) and five types of non-neuronal cells (Müller glia [2720 cells, 12.0%], microglia [2393 cells, 10.5%], retinal pigmented epithelial cells [181 cells, 0.8%], oligodendrocytes [554 cells, 2.4%], and vascular endothelial cells [79 cells, 0.3%]) (Fig. 1b, c). For example, we identified 895 glycinergic amacrine cells, the glycinergic interneurons involved in visual signal transduction[44], which showed strong expression of the solute carrier family 6 member 9 (*slc6a9*), a glycine transporter (Fig. 1c). We found 2720 Müller glia with

apolipoprotein Eb (*apoeb*) expression (Fig. 1c), which is secreted into the vitreous and transported into the optic nerve[45]. Then, we examined the extent to which the retinas of each of the three Wakin goldfish analyzed here contained each cell type. We found that the three samples contained the seven types of neurons and the five types of non-neuronal cells, suggesting the experiment's reproducibility (Fig. 1b, Supplementary Fig. 2). Next, we wondered whether there were any differences in the cell types of the retina between goldfish and zebrafish, a cyprinid teleost species without Cs4R. We compared the cell types of the goldfish retina with those of the zebrafish retina and found the 12 retinal cell types in both species (Supplementary Fig. 3). To confirm whether the composition of each cell-type in the scRNA-seq analysis of goldfish reflects tissue cell composition, we measured cell-type ratios by a different method. The vertebrate retina has three nuclear layers and two plexiform layers where synaptic connections are formed[35]. The outer nuclear layer (ONL) is composed of the photoreceptor bodies, and the inner nuclear layer (INL) is composed of the cell bodies of bipolar, horizontal, and amacrine cells. Meanwhile, the ganglion cell layer (GCL) is mainly composed of ganglion cell bodies. In the vertebrate retina, each cell-type forms a layered structure, and the number of cells in each layer can be counted to determine the approximate number of cell types. Accordingly, we prepared frozen sections from Wakin

retinal tissues, stained with DAPI to stain the nuclei of all retinal cells and counted the number of cells in the different layers of the retina (Supplementary Fig. 3c). The results showed 33.1 ± 6.0% of ONL, 53.6 ± 7.8% of INL, and 13.2 ± 2.1% of GCL ($n = 3$). These values are roughly consistent with the result of our scRNA-seq analysis (29.7% ONL, 54.7% INL, and 1.5% GCL). The number of cells in the GCL in scRNA-seq seems to be smaller than expected from the frozen section experiment. This can be explained because the GCL also contains some amacrine cells. We compared the cell composition of the zebrafish and goldfish retinas (Supplementary Fig. 3d). The numbers in goldfish and zebrafish are roughly similar, however, the most significant difference is the 25.7% rod photoreceptor cells in goldfish compared to only 1.0% in zebrafish. Previous histological studies suggest that the ratio of photoreceptor cells in the retina should be similar between these two species[46]. However, in zebrafish, it is much lower than in tissue sections. Since photoreceptors have long outer segments and connecting cilia, they may be more susceptible to differences in experimental conditions. Differences in filtering and annotating cells during data analysis may also contribute to the differences in the number of rod cells[47]. No major differences were noted among cones, bipolar, and amacrine, suggesting that there were no critical issues in the scRNA-seq analysis performed in this study. This result suggests that the retinal cell population and transcriptomic profile of goldfish and zebrafish are similar, and that the single-cell transcriptome data of the two species are comparable for further subgenome analysis.

**Asymmetric subgenome expression at the cellular level**. We previously performed a bulk RNA-seq analysis in the seven tissues of the goldfish and identified a global gene expression bias toward the L-ohnologs over the S-chromosomes (S-ohnologs) in all tissues analyzed[11,23]. Here, we first tested whether the total gene expression of each cell was biased against the L-subgenome or the S-subgenome using the 11,444 ohnolog pairs. We evaluated the sum of the L-ohnolog and S-ohnolog expression levels for each cell. The sum of the L-ohnolog and that of the S-ohnolog expression levels showed a strong positive and significant correlation (Pearson's correlation coefficient = 0.99, $P < 1.0e-15$, Fig. 2a). This suggests that both the L-ohnologs and S-ohnologs globally contribute to shaping the transcriptome of each cell. The sum of the L-ohnolog expression levels was significantly higher than that of the S-ohnolog expression levels ($P < 1.0e-15$, Wilcoxon test; Fig. 2b). The sum of the L-ohnolog expression levels was higher than that of the S-ohnolog expression levels in 22,365 (98.4%) of the 22,725 cells analyzed (red area in Fig. 2c). This bias in gene expression toward the L- over the S-ohnologs was observed in all 12 retinal cell types (Fig. 2d, left panel). The average ratio of the sum of the L-ohnolog and the S-ohnolog expression levels for each cell type ranged from 1.12 in the retinal ganglion cells to 1.23 in the rod photoreceptor cells (Fig. 2d, right panel). The goldfish genome is still in the process of rediploidization, and both L-ohnologs and S-ohnologs still participate in shaping the transcriptome, as Cs4R is an evolutionarily recent event (~14 MYA)[11]. At the same time, the L-ohnologs and S-ohnologs are in the gradual process of asymmetric subgenome expression at the cellular level[23].

**Ohnolog pairs displaying an expression bias toward the L- or S-ohnolog**. Next, we focused on individual ohnolog pairs. We performed a two-dimensional projection of the dataset by UMAP independently based on the individual gene expression of the 11,444 L-ohnologs or the 11,444 S-ohnologs, respectively. We found that both sets of ohnologs showed similar clusters based on the seven types of neurons and the five types of non-neuronal cells

identified when analyzed using both L-ohnologs and S-ohnologs (Supplementary Fig. 4a). This result indicates that genes on both subgenomes contribute to the characterization of a cell-type-specific transcriptome and are consistent with the finding that ohnologs showed a significant correlation in total gene expression (Fig. 2a). Subsequently, we calculated the average expression levels in all cells for each gene and compared them in all ohnolog pairs. The fold change between the L-ohnolog and S-ohnolog showed a broad distribution, with an average of 1.1 (L-ohnolog/S-ohnolog), indicating a global expression bias toward the L-ohnolog ($P < 1.0e-15$, Wilcoxon test; Supplementary Fig. 4b). In all 12 cell types, we found that the gene expression levels of the L-ohnolog were significantly higher than those of the S-ohnolog (Supplementary Fig. 4b). We plotted the L/S expression ratio for each ohnolog pair in each cell-type (Supplementary Fig. 4c). The L/S peak is centered at zero, suggesting that most ohnolog pairs show no biased expression. This contrasts with the right-shifted peak observed in Fig. 2c (L/S of total gene expression). This result suggests that ohnolog pairs show globally L-biased in terms of total gene expression, but not at most individual gene levels. For each ohnolog pair, we compared the average gene expression between the L-ohnolog and the S-ohnolog in all 22,725 cells. We identified 5123 (44.8%) ohnolog pairs in which the average gene expression in the 22,725 cells was significantly different between the L-ohnolog and the S-ohnolog (Fig. 3a); 2690 (23.5%) ohnolog pairs showed biased expression toward the L-ohnolog over the S-ohnolog, whereas 2433 (21.3%) ohnolog pairs showed biased expression toward the S-ohnolog. The number of ohnolog pairs with biased expression toward the L-ohnolog was significantly higher than that of the ohnolog pairs with biased expression toward the S-ohnolog ($P < 5.0e-4$, Binomial test). Next, we searched for ohnolog pairs among the 11,444 ohnolog pairs that were significantly more highly expressed in one or more specific cell types compared to other cell types (Fig. 3a). We found that 1070 ohnolog pairs (9.3%) showed cell type specific expression profiles (cell-type specificity of the total expression of L- and S-ohnologs) (Fig. 3a). Of these, L-ohnologs showed higher gene expression than S-ohnologs in 260 ohnolog pairs (Fig. 3a, b; Supplementary Data 2), and S-ohnologs showed higher gene expression than L-ohnologs in 245 ohnolog pairs (Fig. 3a, b; Supplementary Data 3). The remaining 10,374 ohnolog pairs (90.7%) showed ubiquitous expression patterns (Fig. 3a). Among them, 2430 ohnolog pairs showed higher gene expression for the L-ohnolog than the S-ohnolog (Fig. 3a, c; Supplementary Data 4), and 2188 ohnolog pairs showed higher gene expression for the S-ohnolog than the L-ohnolog (Fig. 3a, c; Supplementary Data 5). We performed an enrichment test to determine whether L/S-biased genes show more cell-type specific expression than a random set of genes. The result showed no significant difference between them (Fisher's exact test, $P = 0.093$). To characterize the functions of these genes with ubiquitous expression patterns, we performed functional enrichment analysis on these gene sets (Supplementary Fig. 5, 6). In the former gene set (2430 ohnolog pairs), genes related to functions such as neuron development (GO database ID, GO:0048666), cellular response to growth factor stimulus (GO:0071363), and enzyme-linked receptor protein signaling pathway (GO:0007167) were significantly enriched (Supplementary Fig. 5; Supplementary Data 6). In the latter gene set (2188 ohnolog pairs), genes related to functions such as receptor metabolic process (GO:0043112), protein acylation (GO:0043543), and mesenchyme development (GO:0060485) were significantly enriched (Supplementary Fig. 6; Supplementary Data 7).

**Selection of ohnolog pairs retaining unbiased expression at the single-cell-resolution level**. Maintaining gene dosage balance is essential for certain types of ohnolog pairs after WGD[2].

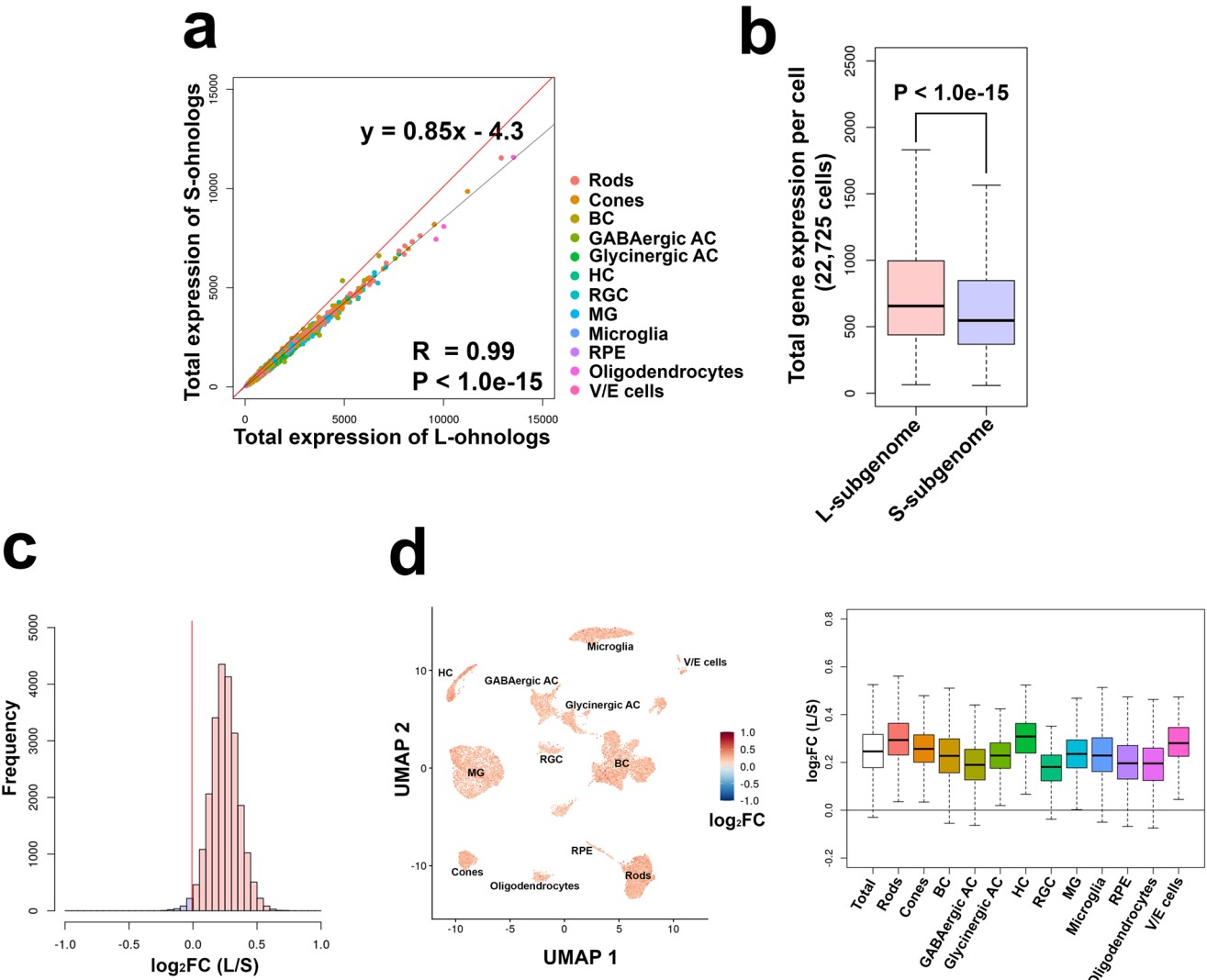

**Fig. 2 Expression bias toward the L-ohnolog over the S-ohnolog in goldfish retinal cells. a** Scatter plot of the total gene expression of L-ohnologs and S-ohnologs in cells. The *x* axis indicates the total gene expression of L-ohnologs and the *y* axis indicates that of S-ohnologs. The dots are colored according to cell types. The total gene expression of L-ohnologs and S-ohnologs showed a positive and significant correlation (Pearson's correlation coefficient = 0.99, *P* < 1.0e–15). The red line indicates *y* = *x*. **b** Boxplots of the total gene expression of L-ohnologs and S-ohnologs in cells. The median of the total gene expression of L-ohnologs was significantly higher than that of S-ohnologs (*P* < 1.0e–15, Wilcoxon test). The ends of the box are the 25 and 75% quantiles. The horizontal line in the box indicates the median. The lines extending from the top and bottom of the box represent the minimum and maximum values. **c** Distribution of the log2-transformed fold changes of the total gene expression of L-ohnologs and that of S-ohnologs in all 22,725 cells. Values higher than 0 (L-ohnolog > S-ohnolog) are colored in red, and values lower than 0 (L-ohnolog < S-ohnolog) are colored in blue. The red line indicates *x* = 0. **d** Distribution of the log2-transformed fold changes of the total gene expression of L-ohnologs and that of S-ohnologs in each cell type. The left panel represents the UMAP plot showing the log2-transformed fold change of the total gene expression of L-ohnologs and that of S-ohnologs in each cell. Cells colored in red indicate that the total gene expression level of L-ohnologs is higher than that of S-ohnolog, whereas the cells colored in blue show the opposite outcome. The right panel represents boxplots of the log2-transformed fold change in the total gene expression of L-ohnologs and S-ohnologs in each cell type. The horizontal line indicates a value of zero. The ends of the box are the 25% and 75% quantiles. The horizontal line in the box indicates the median. The lines extending from the top and bottom of the box represent the minimum and maximum values.

For example, homeobox genes tend to maintain their expression level after WGD because the change in their dosage balance affects embryonic pattern formation and survival[2,48]. To identify ohnolog pairs with unbiased expression, we statistically compared the gene expression between the L-ohnolog and S-ohnolog in all ohnolog pairs for each cell type, and searched for ohnolog pairs in which there was no obvious difference in gene expression between the L-ohnolog and S-ohnolog in any cell type. The results showed that 611 (5.3%) ohnolog pairs had no significant difference in gene expression between the L-ohnolog and S-ohnolog in any cell type (Fig. 4a, Supplementary Data 8). These

genes included several transcriptional factors, such as distal-less homeobox genes (*dlx3b*, *dlx4a*, *dlx4b*, *dlx5a*, and *dlx6a*). In 762 (6.7%) ohnolog pairs, a significant difference in gene expression was found between the L- and S-ohnologs only in one cell type (Fig. 4a, Supplementary Data 9). In the remaining 10,071 (88.0%) ohnolog pairs, the L-ohnolog and S-ohnolog showed significant differences in gene expression in at least two cell types (Fig. 4a). It is suggested that evolutionary constraints acted on the 611 ohnolog pairs without a significant difference in gene expression between the L-ohnolog and S-ohnolog in any cell type. To further obtain biological insights from the list of ohnolog pairs that

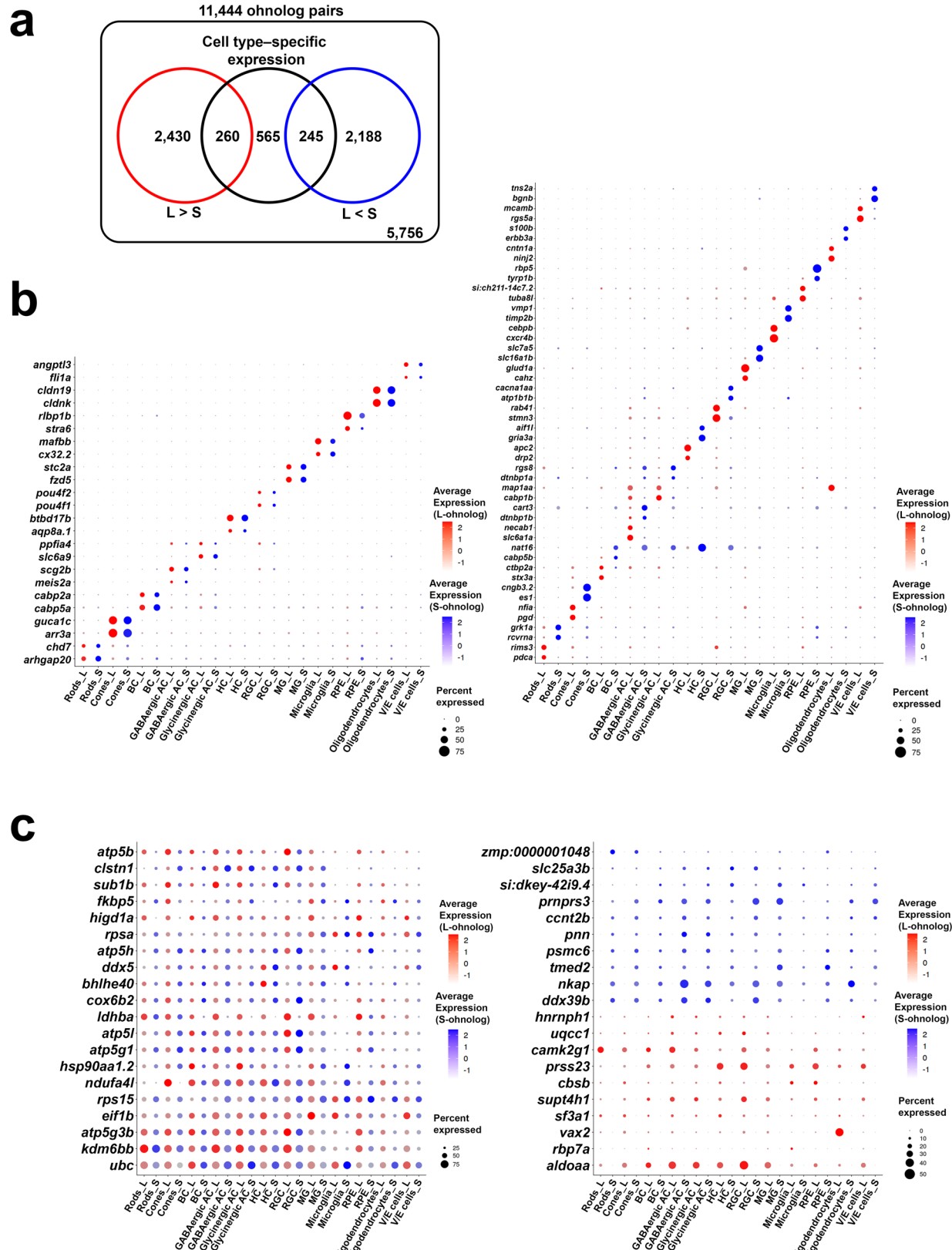

showed no significant difference in gene expression between the L-ohnolog and S-ohnolog in any cell type, we performed a functional enrichment analysis. We found that these ohnolog pairs were significantly related to 295 biological categories, including cytoplasmic ribosomal proteins (WikiPathways ID, WP324), chordate embryonic development (GO:0043009), and

cell fate commitment (GO:0045165) (Fig. 4b; Supplementary Data 10). Some of those 295 biological categories shared similar ohnolog pairs. We generated an enrichment network based on their membership similarities where nodes represented biological categories and edges represented membership similarities with statistical significance. The resulting enrichment network

**Fig. 3 Ohnolog expression profiles among retinal cell types. a** Venn diagram showing the relationship between ohnolog pairs with expression levels that are biased toward the S- or L-ohnolog, and ohnolog pairs with cell-type-specific expression. **b** Ohnologs with cell-type-specific expression. The left panel shows the gene expression of the ohnolog pairs in which both the L-ohnolog and S-ohnolog showed comparable cell-type-specific expression. The right panel shows the gene expression of the ohnolog pairs in which either the L-ohnolog or S-ohnolog showed significantly higher expression than the other. **c** Ohnologs with ubiquitous expression. The left panel shows the ohnolog pairs in which both the L-ohnolog and S-ohnolog are ubiquitously expressed. The right panel shows the gene expression of the ohnolog pairs in which either the L-ohnolog or S-ohnolog showed ubiquitous expression and significantly higher expression than the other.

contained 20 clusters. Notably, we found one large complex cluster consisting of 12 clusters (enclosed clusters, Fig. 4c). The biological categories forming the 12 clusters were composed of several transcription factor genes, including *dlx* genes, homeobox protein NK-2 homolog (*nkx2*) genes, and forkhead box (*fox*) genes. These 12 clusters were broadly related to embryonic development and contained biological categories such as chordate embryonic development (GO:0043009), formation of primary germ layer (GO:0001704), central nervous system development (GO:0007417), and cell fate commitment (GO:0045165).

**Identification of ohnolog pairs showing diversified expression patterns (sub/neo-functionalization).** Previously, we performed bulk RNA-seq on seven goldfish tissues (heart, muscle, bone, brain, eye, gill, and tail fin) and found that 0.46% ohnolog pairs showed sub-functionalization and 3.78% ohnolog pairs showed neo-functionalization[11]. We examined the expression pattern of ohnologs with cell-type-specific expression to identify ohnolog pairs with sub- or neo-functionalization at the single-cell-level resolution. First, we selected the cell type specific ohnologs among the 11,444 ohnolog pairs by searching for ohnologs with significantly higher expression in a particular cell-type (cell-type specificity of individual expression of L- or S-ohnolog). We identified 632 ohnolog pairs in which both the L- and S-ohnologs showed cell-type-specific expression patterns (Fig. 5a; Supplementary Fig. 7, Supplementary Fig. 8; Supplementary Data 11). Next, we compared the expression patterns between the L-ohnologs and the S-ohnologs and divided them into two groups. The first group contained the ohnolog pairs in which both the L- and S-ohnologs exhibited the similar expression patterns (326 ohnolog pairs, 2.8%; Supplementary Fig. 7). The second group included the ohnolog pairs in which the L- and S-ohnologs showed different expression patterns (306 ohnolog pairs, 2.7%; Supplementary Fig. 8). The second group was most likely to contain the sub/neo-functionalized ohnolog pairs (Supplementary Fig. 8). For example, cyclin-dependent kinase 5 regulatory subunit 2b (*cdk5r2b*) is specifically expressed in zebrafish cone photoreceptor cells (Fig. 5a). In the goldfish retina, the L-ohnolog was expressed in the rod photoreceptor cells and the retinal pigmented epithelial cells, in addition to the cone photoreceptor cells, whereas the S-ohnolog of *cdk5r2b* was specifically expressed in the cone photoreceptor cells, as observed in the zebrafish retina (Fig. 5a). This suggests that the L-ohnolog *cdk5r2b* experienced neo-functionalization, whereas its S-ohnolog preserved the original gene function. The solute carrier family 3 member 2a (*slc3a2a*) gene, which encodes an amino acid transporter, was expressed in Müller glia and microglia in the zebrafish retina, whereas the L-ohnolog of this gene was expressed in Müller glia and the S-ohnolog of this gene was expressed in microglia (Fig. 5a). This suggests that the two ohnologs of *slc3a2a* have undergone sub-functionalization. We tested whether the sub/neo-functionalized gene cluster found on scRNA-seq data (306 ohnolog pairs) overlapped with a previously reported sub/neo-functionalized gene cluster found in bulk RNA-seq of tissue (368 ohnolog pairs[11]). This analysis showed that only four genes (*oxr1b*, *rgs16*, *nat16*, and *tubb2*) overlapped in both groups,

indicating that the sub/neo-functionalized genes found by scRNA-seq are significantly different from those obtained by bulk RNA-seq analysis. This indicates that scRNA-seq analysis is useful for identifying sub/neo-functionalized genes. Furthermore, we performed functional enrichment analysis on the 326 ohnolog pairs in which both the L- and S-ohnologs displayed similar expression patterns (Supplementary Fig. 7; Supplementary Fig. 9a; Supplementary Data 12) and the 306 ohnolog pairs in which the L- and S-ohnologs showed different expression patterns (Supplementary Fig. 8; Supplementary Fig. 9b; Supplementary Data 13). We found that both sets of ohnolog pairs were significantly related to biological categories such as cytoplasmic ribosomal proteins (WP324; Supplementary Fig. 9). Notably, the categories of phototransduction (KEGG Pathway, dre04744) or cone photoresponse recovery (GO:0036368) were found in only the 326 ohnolog pairs in which both the L- and S-ohnologs exhibited similar expression patterns (Supplementary Fig. 9). Consistent with this, rhodopsin (*rho*) and opn1lw1 showed non-divergent expression in rods and cones, respectively (Supplementary Fig. 10a). Despite the low expression levels, both L/S-ohnologs of *opn1mw4* and *opn1sw2* are expressed in cones.

We identified 326 ohnolog pairs in which both the L- and S-ohnologs exhibited similar expression patterns (Supplementary Fig. 7); however, these ohnolog pairs may have diversified expression patterns in different subtypes of cells. To address this possibility, we searched for ohnolog pairs showing diversified expression patterns in the UMAP cluster of each retinal cell type. In this analysis, we identified at least four ohnolog pairs, including calcium binding protein 5a (*cabp5a*), 5b (*cabp5b*), retinoschisin 1a (*rs1a*), and prostaglandin-endoperoxide synthase 2b (*ptgs2b*) that showed diversified expression patterns in the UMAP clusters (Fig. 5b and Supplementary Fig. 10b). For example, *cabp5aL* expression was enriched in certain subpopulations of the bipolar cell cluster (red arrowheads in Fig. 5b), whereas *cabp5aS* expression was enriched in the different subpopulations of the bipolar cell cluster (blue arrowheads in Fig. 5b). These results show that ohnologs are differentially expressed even at the subtype level of retinal cells.

**Determination of the open chromatin regions (OCRs) of individual cells by scATAC-seq.** Chromatin accessibility in the promotor regions dynamically controls gene expression[49]. Therefore, we wondered whether a divergent evolution of the L-ohnologs and S-ohnolog is also observed in the accessibility of promoter regions in single-cell resolution using single-cell Assay for Transposase-Accessible Chromatin sequencing (scATAC-seq). The scATAC-seq technology profiles the accessible chromatin regions using a genetically engineered hyperactive DNA transposase (Tn5) that cleaves and tags OCRs with single-cell resolution[50]. We performed scATAC-seq on the retina of a single Wakin individual and mapped OCRs in 19,750 cells. We identified 245,817 OCRs across the genome, of which 132,681 (54.0%) were located in the L-subgenome and 113,136 (46.0%) were located on the S-subgenome (Supplementary Data 14). To comprehensively quantify the promoter accessibility of the genes in each cell, we counted the number of reads from OCRs located on

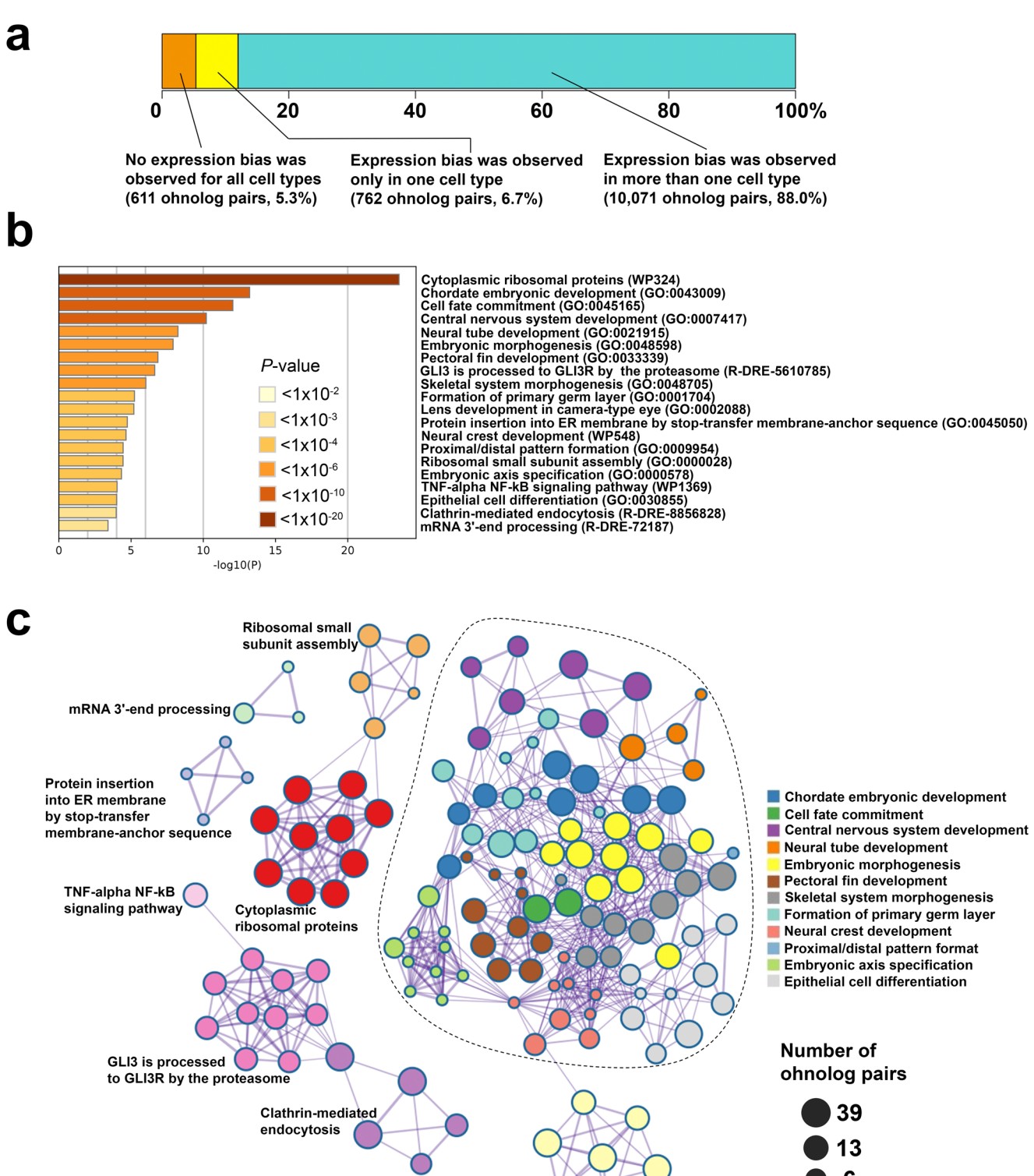

**Fig. 4 Functional enrichment analysis of the ohnolog pairs with retained unbiased expression. a** Of the 11,444 ohnolog pairs analyzed, 611 (5.3%) showed retained unbiased expression in each cell type (orange). For the remainder of the ohnolog pairs, expression biases toward one of the ohnologs were detected in one (yellow) or more (blue) cell types. **b** Functional enrichment analysis of the 611 ohnolog pairs that showed retained unbiased expression in each cell type. The *x* axis represents the negative log10-transformed *P* value based on the accumulative hypergeometric distribution[67]. The *y* axis represents the 20 biological categories. Deeper color of the bar plot means smaller *P* value. **c** Network representation of the statistically enriched categories in the functional enrichment analysis of the ohnolog pairs with retention of unbiased expression. The nodes represent the enriched categories and the edges are defined based on the similarities among their gene memberships. The name of the cluster is adopted from the name of the cluster with the smallest *P* value among the biological categories contained in that cluster. The dotted line highlights the large complex cluster consisting of 12 clusters. The node size is proportional to the number of input ohnolog pairs grouped into each category.

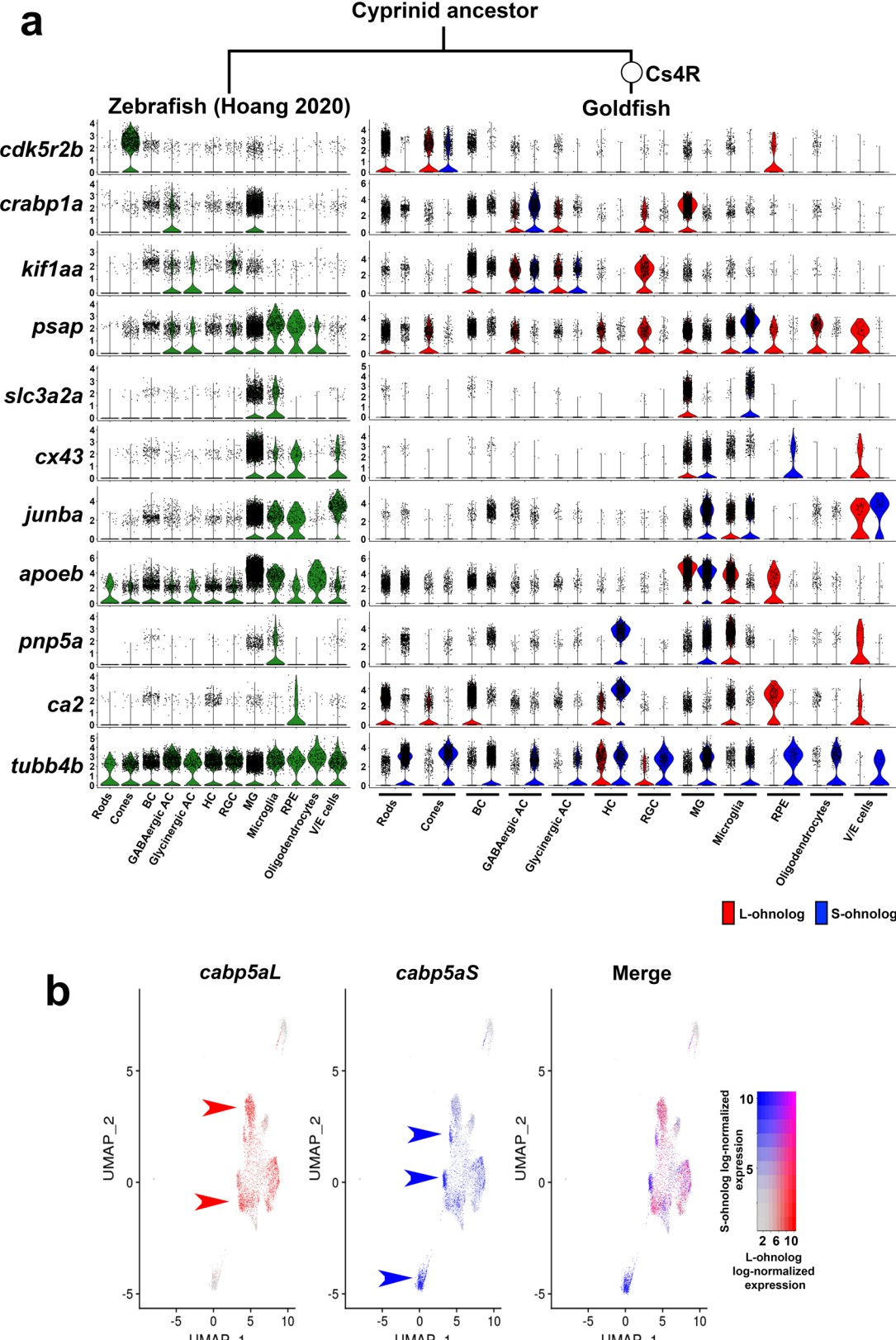

**Fig. 5 Ohnolog pairs in which the L-ohnolog and S-ohnolog show different expression profiles. a** Violin plots showing the gene expression of ohnolog pairs in which the L-ohnolog (red) and S-ohnolog (blue) exhibit different expression profiles. The scRNA-seq data of the zebrafish retina[42] are shown on the left, for reference. **b** The *cabp5a* ohnologs exhibit a diversified expression pattern in the bipolar cells of the UMAP cluster. The red or blue arrowheads indicate the enriched cells expressing *cabp5aL* or *cabp5aS,* respectively.

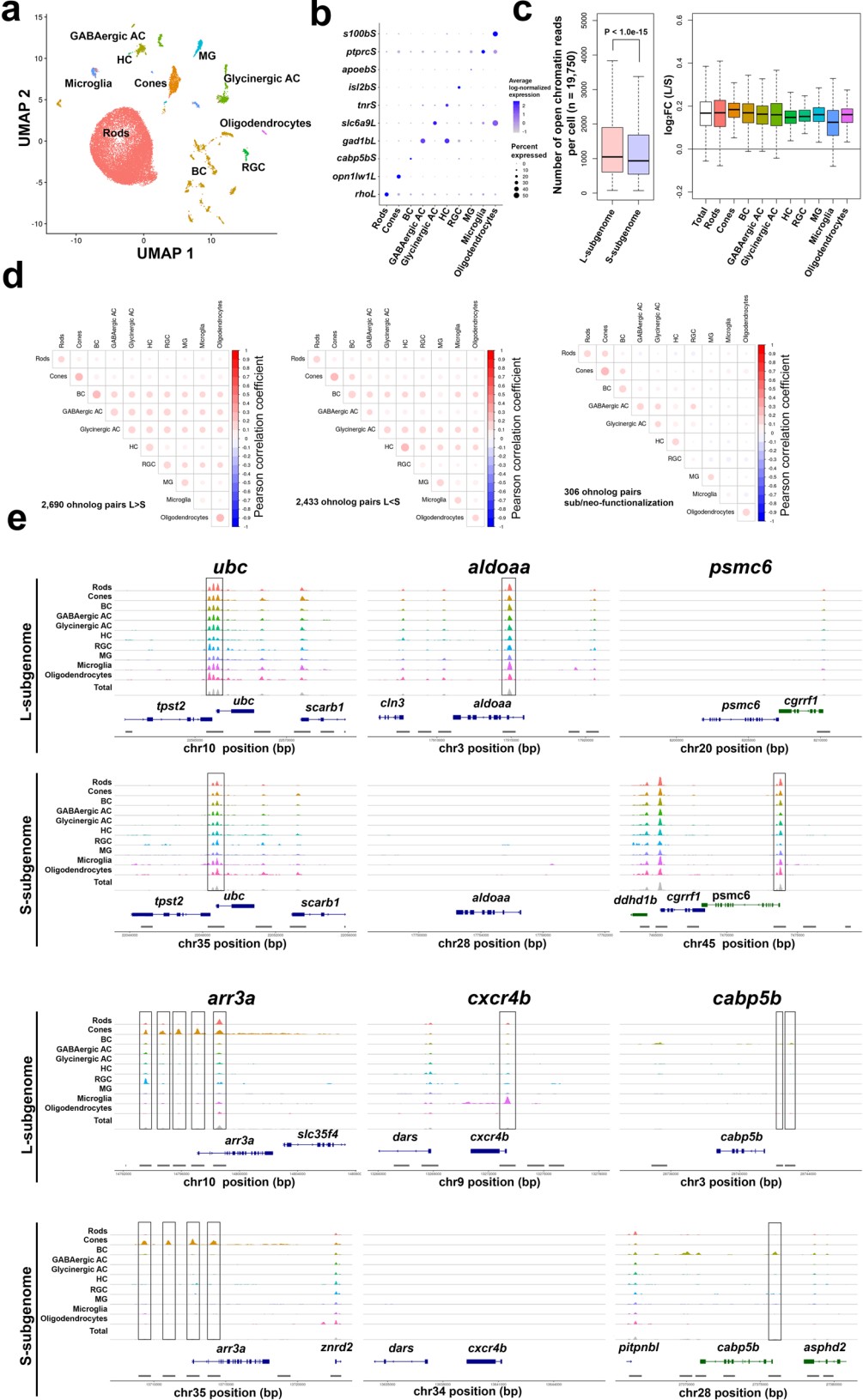

the gene body and the region up to 2 kb upstream of the transcriptional start site in each of the 11,444 ohnolog pairs (22,888 genes). We identified the seven types of neurons that were also identified by the scRNA-seq analysis, and the three types of non-neuronal cells, including Müller glia, microglia, and oligodendrocytes (Fig. 6a). These ten cell types had OCRs in the promotor regions of the known cell-type marker genes (Fig. 6b). The promotor regions of L-ohnologs showed higher overall accessibility than those of S-ohnologs in all cell types (Fig. 6c, left panel). Moreover, for each cell type, the promotor regions of L-ohnologs showed higher overall accessibility than those of S-ohnologs (Fig. 6c, right panel).

**Fig. 6 scATAC-seq analysis of the goldfish retina. a** UMAP plot showing the cellular composition of the Wakin goldfish retina. The 19,750 cells were projected into a two-dimensional space by UMAP. **b** Promoter accessibility of the cell-type-specific marker genes. **c** Promoter accessibility of ohnologs. The left panel shows boxplots of the total number of open chromatin reads in the promotor regions in each cell. The right panel shows boxplots of the log2-transformed fold change of the total number of open chromatin reads in the promotor regions of L-ohnologs and S-ohnologs in each cell type. The ends of the box are the 25 and 75% quantiles. The horizontal line in the box indicates the median. The lines extending from the top and bottom of the box represent the minimum and maximum values. **d** Pearson correlation coefficient matrix between observed cell types for each of the gene sets using the data from scRNA-seq and scATAC-seq analysis. The correlations between scRNA-seq and scATAC-seq are significantly high between the corresponding cell types for any gene set (left panel, 2690 ohnolog pairs showed biased expression toward the L-ohnolog over the S-ohnolog; middle panel, 2433 ohnolog pairs showed biased expression toward the S-ohnolog over the L-ohnolog; right panel, 306 ohnolog pairs with sub/neo-functionalization). **e** Accessible chromatin landscape of each cell type. The x axes represent chromosomal positions (bp) and the y axes represent chromatin accessibilities. The peaks of scATAC-seq are highlighted by the black lines. The genome annotation is based on the NCBI Gnomon genome annotation of RefSeq goldfish genome assembly (GCA_003368295.1).

Then, we focused on the potential regulatory regions underlying the expression patterns found on scRNA-seq analysis. We compared significant peaks of OCRs in the scATAC-seq data with mRNA expression patterns on scRNA-seq data. To evaluate the relationship between these two methods, we calculated a Pearson correlation coefficient matrix between the observed cell types for each gene set. We found significantly high correlations between the corresponding cell types for any gene set in scRNA-seq and scATAC-seq (Fig. 6d), suggesting that the gene expression profiles in retinal cell types from our scRNA-seq analysis and the OCR profiles in retinal cell types scATAC-seq data have essentially similar features.

Using these data, we predicted the candidate regulatory regions of the bias-expressed genes (L-bias 2690 ohnolog pairs, 6083 OCRs, Supplementary Data 15; S-bias 2443 ohnolog pairs, 5810 OCRs, Supplementary Data 16). These data provide a comprehensive view of the evolution of potential gene regulatory regions after WGD of the goldfish ancestor genome. Subsequently, we evaluated the accessibility of the gene bodies and promoter regions of the genes that showed distinct expression patterns in scRNA-seq. We found that the accessibility of the gene bodies and promotor regions exhibited a pattern that was similar to that of genes with distinct expression patterns measured by scRNA-seq (Fig. 6e). For example, both ohnologs of ubiquitin c (*ubc*) were ubiquitously expressed in scRNA-seq (Fig. 3c, left panel), and the promoter accessibility of both ohnologs showed a similar pattern (Fig. 6e). The L-ohnolog of aldolase, fructose-bisphosphate aa (*aldoaa*) was ubiquitously expressed (Fig. 3c, right panel), and the S-ohnolog of proteasome 26 S subunit ATPase 6 (*psmc6*) was ubiquitously expressed (Fig. 3c, right panel), and a similar bias was observed in the promoter accessibility (Fig. 6e). The arrestin 3a (*arr3a*), chemokine (*C-X-C* motif) receptor 4b (*cxcr4b*), and calcium-binding protein 5b (*cabp5b*) genes showed distinct expression patterns in cone photoreceptors, microglia, and bipolar cells, respectively (Fig. 3b), as well as different promoter accessibility (Fig. 6e). These results suggest that the bias toward L-ohnologs over S-ohnologs and distinct expression patterns observed in scRNA-seq are also present in promoter accessibility at the individual cell level.

Next, we focused on the evolution of regulatory regions for cell-type-specific ohnolog pairs. We identified OCRs for sub/neo-functionalization genes (306 ohnolog pairs, 849 OCRs, Supplementary Data 17). These data sets are therefore candidates for cell type-specific, subgenome-evolved regulatory regions. We further used this data set to search for Otx2/Crx binding sequences (Fig. 7a) associated with photoreceptor/bipolar cell-specific gene expression in the retina. Otx2 and Crx are essential transcription factors for retinal development and maintenance[51]. This analysis revealed that Otx2/Crx binding sequences are significantly enriched in the OCRs of these cell types ($P < 1.0e-15$, Wilcoxon test, Fig. 7b). We described 89 OCRs containing Otx2/Crx binding sites for regulation of ohnolog pairs with photoreceptor/bipolar cell-specific expression (Supplementary Data 18). These OCRs are candidate regulatory regions for cell type specificity of differentially expressed ohnolog pairs between L/S-subgenomes. For example, scRNA-seq analysis showed that *prph2b*, a photoreceptor specific glycoprotein, is more highly expressed in the L-subgenome than in the S-subgenome (Fig. 7c). Consistent with this, scATAC-seq analysis revealed significant OCRs with Otx2/Crx binding sites in the L-subgenome but not in the S-subgenome (Fig. 7c). Accordingly, our scATAC-seq data provide information on gene regulatory regions important for the evolution of gene expression in retinal ohnolog pairs.

## Discussion

The recent development in scRNA-seq analysis offers unprecedented qualitative advantages over analysis of gene expression profiles at the tissue or organ levels (bulk RNA-seq). New cell types have been discovered in the developing and mature tissues as well as in tumors and organoids, and their gene expression profiles determined[37,52–54]. The study of gene evolution after WGD using organisms that have undergone a recent WGD is one of the most desired areas for single-cell analysis, as it is difficult to distinguish between substantive expression differences in ohnologs and averaged expression in ohnologs because of the presence of several types of cells in a specific tissue. To overcome such limitations, this study used scRNA-seq and scATAC-seq to analyze the evolution in expression profiles of ohnologs in the retinal tissues of allotetraploid goldfish. As a result, diversified evolution in gene expression profiles of ohnolog pairs were observed at single-cell resolution after recent WGD (Supplementary Fig. 11).

Firstly, using the gene expression data sets of 11,444 ohnolog pairs to screen 22,725 retinal cells identified 12 types of retinal cells. A higher sum (98.4%) of expression levels was observed in L-ohnolog than in S-ohnolog. Similar bias in gene expression was observed in all the 12 retinal cell types examined. These results suggest that L-subgenome is the dominant in goldfish retinal cells, and demonstrate that asymmetric evolution in gene expression occurred at individual cell level in the goldfish retina. With respect to asymmetric subgenome evolution, previous studies have reported a positive correlation of sequence similarity between ohnologs and similarity in ohnolog expression patterns[11]. Plant genome analysis propose a trade-off hypothesis (a trade-off between reduced transposition and deleterious effects on neighboring gene expression) of asymmetric subgenome evolution in which epigenetic silencing of transposable elements regulate gene expressions[20,55]. Despite the unknown mechanism

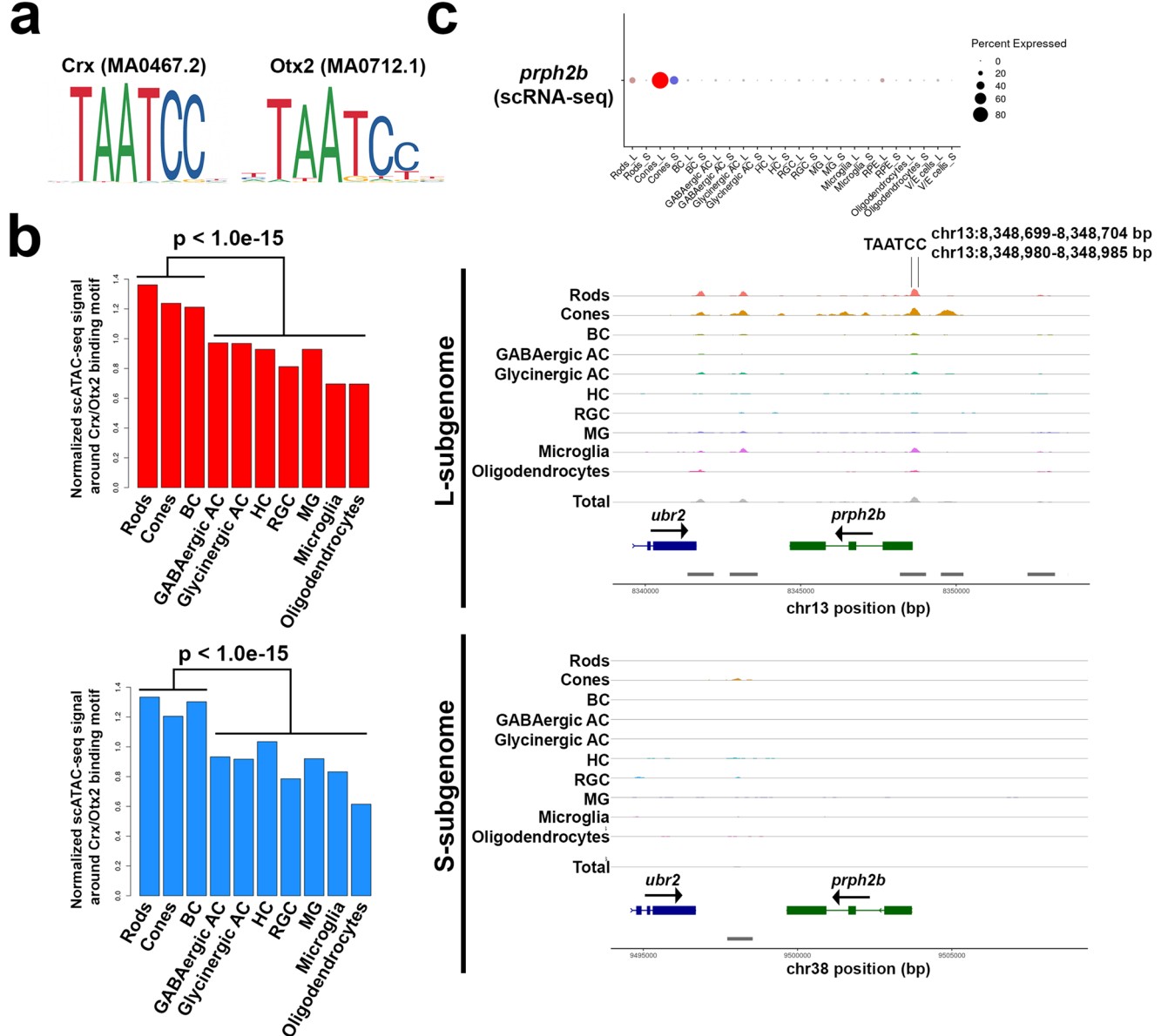

**Fig. 7 Analysis of Otx2/Crx binding sequence based on scATAC-seq. a** Binding motif of Crx and Otx2. IDs in parentheses come from JASPAR[68]. **b** Median value of normalized scATAC-seq signals around the Crx/Otx2 binding motif in each OCR is higher in rods, cones, and BC than in other cell types. Upper panel is for the L-subgenome and the lower is for the S-subgenome. **c** Expression pattern of the *prph2b* ohnolog pair (example of L-subgenome dominant ohnolog pair) based on scRNA-seq analysis data (upper panel). OCR peaks of the *prph2b* ohnolog pair are based on scATAC-seq data (lower panel). The expression data from scRNA-seq analysis is consistent with the results of scATAC-seq analysis. The genome annotation is based on the NCBI Gnomon genome annotation of RefSeq goldfish genome assembly (GCA_003368295.1).

of asymmetric subgenome evolution observed in vertebrates including African clawed frogs, it would be interesting to determine whether the observed asymmetric subgenome evolution is consistent with this hypothesis.

Secondly, our functional enrichment analysis revealed that the 2430 ohnolog pairs showing the L-ohnolog-biased expression were associated with neuron development, whereas the 2188 ohnolog pairs with the S-ohnolog-biased expression were associated with mesenchyme development. A previous bulk RNA-seq analysis of goldfish tissues showed a more frequent asymmetric changes in gene expression toward the L-ohnologs in the brain or eye than in the heart or muscle[23]. These results could be due to the fact that L-ohnologs and S-ohnologs are relatively more functionally dominant in neuronal and mesenchymal cells in goldfish, respectively.

We also identified 611 (5.3%) ohnolog pairs with no obvious difference in gene expression between the L-ohnolog and S-ohnolog in all cell types. These ohnolog pairs could be dosage-sensitive genes, and a loss in their dosage balance after WGD would have been detrimental to survival[2]. A previous bulk RNA-seq analysis of seven goldfish tissues revealed that ~70% of the ohnolog pairs exhibited coexpression in at least three tissues[11], suggesting that single-cell transcriptome analysis is more efficient in identifying dosage-sensitive genes. In addition, these dosage-sensitive ohnolog pairs were found to be significantly associated with biological processes, including chordate embryonic development and cell fate commitment. This is consistent with findings that genes encoding transcription factors and signal transducers of development have been preferentially retained after WGD in plants and vertebrates[1,2]. The conversion of these

ohnolog pairs to singletons is probably restricted, and their expression levels remain evolutionary maintained in the goldfish genome.

Ohnolog pairs with sub/neo-functionalization are often associated with the diversified tissues among species over evolution. The evolutionary diversity associated with visual functions, such as wavelength shift, photoreceptor distribution, interneuron responses, and retinal layer structure, is widely observed in vertebrates[29–31]. Sub/neo-functionalization of ohnolog pairs after WGD in photoreceptor evolution has been observed in teleost fish[56,57]. In addition, such evolutionary changes are not only limited to photoreceptor cells, but also might occur in interneurons, including bipolar cells, horizontal cells, amacrine cells, and ganglion cells[58].

We identified 306 ohnolog pairs with diversified expression patterns that likely harbored sub/neo-functionalized ohnologs. Functional enrichment analysis found that these ohnolog pairs were significantly associated with biological processes, such as calcineurin and calcium regulation, dopamine secretion, and transmission across chemical synapses. These biological processes are related to synapse function, suggesting that the ohnolog pairs tend to undergo sub/neo-functionalization after Cs4R, leading to functional diversification of retinal cells. These results also suggest that the observed diversity in ohnolog expression in the goldfish retina might have contributed to their visual evolution during the early stage after Cs4R to adapt to the environment using these biological processes. Notably, ohnolog pairs with phototransduction functions did not group together with the sub/neo-functionalized members but clustered with other ohnolog pairs having similar expression pattern. This finding suggests that the evolution of these phototransduction ohnologs is unlikely to deviate from photoreceptor-specific expression patterns, and that photoreceptor-specific genes are more likely to evolve through other mechanisms such as post-transcriptional modification or amino acid mutation. It is known that most rod and cone phototransduction-related genes (e.g., CNGA1 vs CNGA3) were generated on 1 R/2 R WGD[59]. Therefore, additional sub/neo-functionalization of rod and cone phototransduction-related genes after Cs4R seems unlikely. Although this study focused primarily on the expression levels of mRNA ohnolog pairs, it is difficult to completely rule out that reduced gene expression does not result in reduced protein levels or loss of gene function in a particular cell-type. However, because ohnolog pairs show highly similar sequences, it is likely that their protein stability and/or activity are similar in the cells. Thus, in most cases, the mRNA expression levels of ohnolog pairs will most likely reflect protein amount and/or activity.

We analyzed ohnolog pairs that acquired diversified expression among the 12 retinal cell types, and identified at least four ohnolog pairs, including cabp5a, cabp5b, rs1a, and ptgs2b, that showed diversified expression in bipolar cells and Müller glia (Fig. 5, Supplementary Fig. 10). Notably, the expression of cabp5aL was enriched in certain subpopulations of the bipolar cell cluster, whereas the expression of cabp5aS was enriched in different subpopulations of the bipolar cell cluster. Cabp5 is a pan-bipolar cell marker[60] that affects the sensitivity of light response by modulating the transmission of light signals through the retinal circuitry[61]. These results suggest the possibility that the goldfish ancestor acquired diversified sensitivity to light responses, such as specific wavelength (cone photoreceptor functions) and/or dim light (rod photoreceptor functions) after developing Cs4R to adapt to the environment via sub/neo-functionalization of ohnolog pairs. However, future functional analysis of ohnologs is necessary to confirm this hypothesis. The observed expression patterns of sub/neo-functionalized ohnologs in the goldfish retina

suggested that evolution in their expression occurred at a relatively early stage after WGD.

Thus, our results indicate that single-cell transcriptome analysis of the retina is a suitable technique for studying gene evolution after a recent WGD. Similar single-cell analysis of the retina in the salmonid fish[62], and the African clawed frog which underwent WGD 80 MYA, and 18 MYA[18], respectively would also provide additional valuable information. The comparison of scRNA-seq data from these organisms will improve our understanding of the early stage of gene evolution after WGD. In addition, we showed that combining both scRNA-seq and scATAC-seq analysis of the retina is effective for studying gene expression differences between subgenomes, and can reliably be used to analyze organisms that have recently undergone WGD.

## Methods

**Animal care**. All goldfish in this study were bred in Japan. Wakin goldfish were maintained at Nagahama Institute of Bio-Science and Technology. The water in the adult goldfish tanks was changed on a monthly basis.

**Study approval**. All procedures were reviewed and approved by the Institutional Safety Committee on Recombinant DNA Experiments (approval ID 0849) and the Animal Experimental Committees (approval ID 087) at Nagahama Institute of Bio-Science and Technology and were performed in compliance with institutional guidelines.

**Preparation of single-cell suspensions for scRNA-seq and scATAC-seq**. The goldfish were anesthetized with tricaine (100 mg/L). The eyes from adult goldfish were enucleated and the retinas were dissected in cold phosphate-buffered saline (PBS) under a stereo microscope (S9, Leica). The retinal tissues were incubated in 1 mL of enzyme solution (Neuron dissociation solutions 291–306, Fujifilm) at 32 °C for 5 min. Cells were dissociated by repeated gentle pipetting (10 times) and incubated at 32 °C for 5 min. Then, cells were pelleted by centrifugation at 4 °C for 5 min at 1800 rpm. The supernatant was carefully removed, 1 mL of disperse solution (Neuron dissociation solutions, Fujifilm) was added, and cells were resuspended by repeated gentle pipetting (10 times). Cells were passed through a 40-μm cell strainer to remove cellular aggregates. Cell viability was examined and confirmed to be 80% or more by Trypan Blue staining.

**scRNA-seq and scATAC-seq library construction and sequencing**. For scRNA-seq, single-cell suspensions were loaded onto the 10x Genomics Chromium Single Cell system using the v3.1 chemistry per manufacturer's instructions. Approximately 16,500 live cells per sample were loaded to obtain the transcriptomes. For scATAC-seq, the cells were washed once in ice-cold PBS and lysed in ATAC lysis buffer (10 mM Tris-HCl pH 7.4, 10 mM NaCl, 3 mM MgCl$_2$, 0.1% Tween-20, 0.1% Nonidet P40, 0.01% Digitonin, 1% BSA). After 5 min incubation on ice, the nuclei were washed with ATAC wash buffer (10 mM Tris-HCl pH 7.4, 10 mM NaCl, 3 mM MgCl$_2$, 0.1% Tween-20, 1% BSA). The nuclei were strained using 40-μm Flowmi strainers (Sigma-Aldrich) and counted on CellDrop (Devinox) by AOPI staining. Transposition and single nucleus ATAC-seq library preparation were conducted by using Chromium Next GEM Single Cell ATAC v1.1 Reagent (10x Genomics) according to manufacturer's protocols. The DNA concentration of the libraries was measured on an Agilent 2100 Bioanalyzer (Agilent Technologies) and sequenced using Hiseq 2500 (Illumina).

**Generation of gene expression matrices**. The goldfish reference genome sequence (GCA_014332655.1) was downloaded from the NCBI Assembly database. We annotated this assembly by mapping the NCBI Gnomon genome annotation of RefSeq goldfish genome assembly (GCA_003368295.1) to it using the Liftoff tool[63]. We indexed the goldfish reference transcriptome using the cellranger mkref command for scRNA-seq analysis (provided by 10x Genomics). The goldfish reference genome was indexed by cellranger-atac mkref command for scATAC-seq analysis. The sequencing reads of scRNA-seq were mapped to the indexed goldfish reference transcriptome and unique molecular identifiers (UMI) counts were generated using the cellranger count command with default parameters. The sequencing reads of scATAC-seq were mapped to the indexed goldfish reference genome and UMI counts were generated using cellranger-atac count command with default parameters.

**Definition of ohnolog pairs**. We previously reported a genome assembly of the Wakin goldfish genome[11]. This assembly contains heterogeneous diploid regions in approximately 22% of the genome[11]. When redundant sequences are included in the genome assembly, it is difficult to align the sequencing reads of scRNA-seq on such regions, and correction is required in the subsequent gene expression quantification. To overcome this, we mapped the NCBI RefSeq annotation of our Wakin

genome assembly, which contains 53,052 protein-coding genes, to another recently reported assembly of the goldfish genome generated by Hi-C scaffolding[25]. Of the 53,052 protein-coding genes, 44,104 protein-coding genes (83.1%) were mapped on the 50 chromosomes, of which 23,438 genes were mapped to L-chromosomes and 20,666 genes to S-chromosomes. We further performed reciprocal BLAST analysis between the 23,438 genes on the L-chromosomes and the 20,666 genes on the S-chromosomes, and identified reciprocal best hits. We defined ohnolog pairs as those simultaneously satisfying the following two conditions: (1) both ohnologs must be present in the homeologous chromosome pair of goldfish, and (2) only one zebrafish ortholog can be identified in the zebrafish chromosome orthologous to the goldfish chromosomes on which the ohnolog pair is localized. This analysis rendered 11,444 ohnolog pairs with a high degree of confidence (Supplementary Fig. 1b, c; Supplementary Data 1). Using MCScanX[64], we tested whether the 11,444 ohnolog pairs formed colinear synteny blocks, finding that 97.1% of the ohnolog pairs formed statistically significant colinear synteny blocks (Supplementary Fig. 1d–f). Most goldfish specific small-scale duplicates were excluded in this strategy as they are expected to be localized to nonhomeologous chromosomes.

**Quality control and data preprocessing**. For scRNA-seq analysis, the UMI matrices were used as input for the Seurat v4 package[65]. For each UMI matrix from each sample, we generated the Seurat class object and performed downstream analyses. We normalized the gene expression for each cell by the total gene expression, multiplied this by a scale factor set to 10,000. We then performed natural-log transformation using the NormalizeData method implemented in Seurat. We scaled the expression of each gene using the ScaleData method implemented in Seurat so that the average gene expression across cells is 0 and the gene expression variance across cells is 1.

For scATAC-seq analysis, we used Signac, a framework for scATAC-seq analysis, an extension of the Seurat v4[66]. To generate a gene activity matrix, we extracted the gene coordinates from the goldfish genome annotation and extended them to include the 2 kb upstream region followed by counting the number of fragments for each cell that map to each of these regions using the GeneActivity method. We then normalized and scaled the gene activity matrix using the NormalizeData method. We searched for the Otx2/Crx binding motif (TAATCC) in the genome sequence and defined the regions extending from 100 bp upstream to 100 bp downstream of Otx2/Crx binding sequences. For each cell-type, we compared the average expression value in scRNA-seq and the average scATAC-seq signals for each gene.

**Dimensionality reduction**. To project scRNA-seq data into a low dimensional space, we performed principal component analysis (PCA) using the RunPCA method implemented in Seurat. We then performed the UMAP based on the first 40 principal components using the RunUMAP method. For dimensionality reduction of scATAC-seq data, we performed latent semantic indexing (LSI) using the RunSVD method. Then, we performed UMAP based on the second to 30 highest components of LSI using the RunUMAP method.

**Cell subpopulation identification**. We identified ohnolog pairs by reciprocal BLAST best-hit method with an $E$ value < 1.0e–40. Cells were subjected to graph-based clustering by the combination of the FindNeighbors and FindClusters methods. By referring to the expression of well-known cell type marker genes, we assigned a cell type identity to each cell cluster. To find specifically expressed genes in each cell type, we used the FindAllMarkers method.

**DAPI staining of the Wakin retinal section**. We prepared frozen sections from Wakin retinal tissue. The goldfish were anesthetized with tricaine. The eyes from adult goldfish ($n = 3$) were enucleated and the retinas were dissected in PBS. The retina were fixed in 4% paraformaldehyde in PBS for 3 h at room temperature. Fixed retina were cryoprotected using 30% sucrose in PBS, embedded in Tissue-TekOCT compound 4583 (Sakura), frozen, and sectioned. Sections were dried, rehydrated in PBS, incubated in blocking solution (5% normal goat serum and 0.5% Triton X-100 in PBS) for 1 h with DAPI (1 μg/ml). Slides were washed with PBS three times. Specimens were observed using a fluorescent microscope. The number of cells were counted in different layers of the retina.

**Identification of differentially expressed genes**. To identify differently expressed genes, we performed Wilcoxon tests and calculated adjusted $P$ values based on Bonferroni correction for all genes using the FindMarkers method implemented in Seurat. Genes with an adjusted $P$ value < 0.05 and with more than twofold change were regarded as differently expressed.

**Functional enrichment analysis**. For functional enrichment analysis, the gene IDs of zebrafish orthologs were uploaded to the Metascape server[67]. For each given gene list, functional enrichment analysis was performed with the following ontology sources: GO Biological Processes, KEGG Pathway, Reactome Gene Sets, and WikiPathways. All functional enrichment analyses were performed with default parameters.

**Statistics and reproducibility**. All statistical analysis were conducted with a significance level of $P < 0.05$. Experiments were repeated in at least two independent preparations to confirm reproducibility of the results. We analyzed scRNA-seq data of total 22,725 cells from Wakin goldfish ($n = 3$, biologically independent samples). The analyzed cell numbers of each cell type are as follows: rod photoreceptor cells, 5842 cells; cone photoreceptor cells, 911 cells; bipolar cells, 7080 cells; GABAergic amacrine cells 1064 cells; glycinergic amacrine cells, 895 cells; horizontal cells, 668 cells; and retinal ganglion cells, 338 cells; Müller glia, 2720 cells; microglia, 2393 cells; retinal pigmented epithelial cells, 181 cells; oligodendrocytes, 554 cells; vascular endothelial cells, 79 cells. We analyzed three sections from three Wakin goldfish individuals for DAPI staining. All statistical analysis including calculation of Pearson's correlation coefficient and Wilcoxon test, Fisher's exact test, and Binomial test were conducted in the R software environment version 4.1.2.

**Reporting summary**. Further information on research design is available in the Nature Portfolio Reporting Summary linked to this article.

## Data availability

The sequencing data of scRNA-seq and scATAC-seq has been deposited in DDBJ Sequence Read Archive under accession number PRJDB12920. Source data underlying the figures presented in this study are available at Figshare (https://doi.org/10.6084/m9.figshare.20496825).

## Code availability

The scripts used for analyses in the current study are available at Zenodo (https://doi.org/10.5281/zenodo.7387954).

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

## Acknowledgements

This work was supported by Grant-in-Aid for Scientific Research (19H03420, 22H02825, and 19K22426 to Y.O.) from the Japan Society for the Promotion of Science, Platform for Advantaged Genomics (JSPS KAKENHI, 16H06279 PAGS to Y.O.), NIG-JOINT (24A2020 and 20A2021 to Y.O.), the Mishima Kaiun Memorial Foundation to Y.O., the Naito Science & Engineering Foundation to Y.O., Takeda Science Foundation to Y.O. and T.K., the Mochida Memorial Foundation for Medical and Pharmaceutical Research to T.K. and the International Medical Research Foundation to T.K.

## Author contributions

T.K. and Y.O. designed the study. Y.O. supervised the project. T.K., K.S., M.I., H.T., A.T., and Y.O. conducted laboratory experiments. T.K., K.F., Z.C., K.K., M.I., S.M.B., H.N., A.T., and Y.O. performed the bioinformatic analysis. T.K. and Y.O. mainly wrote the manuscript. All authors provided feedback on the manuscript.

## Competing interests

The authors declare no competing interests.
