## [Peer Review File · Communications Biology]

Reviewers' comments:

Reviewer #1 (Remarks to the Author):

The authors conducted an intensive transcriptome analysis in the goldfish retina. Their data of transcriptome analyses also carefully examined by using sophisticated statistic analysis. Based on these analyses, the authors provided the interpretation of how the expression levels of the duplicated genes have changed during evolutionary course. The data they collect would be useful for other researchers and should be published. However, I would like to make a few suggestions to the authors regarding the consistency of the introduction and the discussion as well as generalities of their findings. More specifically said, from this manuscript, it is unclear why the authors motivated for sticking to the combination of goldfish, retina and whole genome duplication. The authors should elaborate more on the biological or evolutionary significance of their focus. I indicated several difficult-to-understand points (Q1-Q4) in this manuscript (please see below).

Introduction:

Q1:

Page 4, para 1, "Therefore, determining the expression levels of orthologues in a single cell..."

I assumed the authors aimed to narrow down their research focus from the first paragraph to second paragraph on Page 4. However, the contextual flow seems to be interrupted by the above indicated sentence.

Q2:

Page 4, para 2 "Therefore, WGD is most likely to contribute to the evolution of retinal diversification in vertebrates, however, a detailed analysis on the process of gene evolution during the early stages after WGD remains unclear"

The context more diverged in the second paragraph on page 4. The authors raised different issues in this paragraph. They explained the specific feature of the retina of vertebrates as a light-sensitive tissue and the two rounds of WGD which occurred in the common ancestor of the vertebrates. My understanding is that the authors want to investigate how recently duplicated genes evolved at the level of gene expression patterns by using retina; I hope my understanding is correct. However, to me, it was unclear why the authors had to use the retina, rather than the other tissues. The retina is not the sole diversified tissues in the lineage of the vertebrate relating with WGD. Sophisticated skeletons, brains, and etc. are also known as the diversified tissues and organs in the vertebrate lineages. I recommend that the author more carefully explain their motivation why they focus on the retina.

Q3 (continued from the above)

Discussion

Page 12 para 4 "This is consistent with findings that..."

Page 13, para 1 "The evolutionary diversity associated with visual functions..."

The authors explained their finding about the analyses. On line 341, they described that their results are consistent with previous studies on plants and vertebrates, but the next paragraph simply explained the visual functions of vertebrates. Although I know that the authors use the retina for their research, it is hard for me to understand how the former and the latter paragraphs are connected.

I strongly recommend that the authors consider this discussion part, taking into consideration the above comment (Q2).

Q4.

Page 13, para 2 "We identified 306 ortholog pairs..."

From this paragraph, the finding of the diversified expression patterns is described. However, from these descriptions, it is hard to distinguish their finding is the goldfish specific evolutionary events or comparable with the other WGD events. Although the statements about four genes (cabp5a, cabp5b, rs1a, and ptgs2b) are relatively specific in the next paragraph (line 363), the statement about 306 ortholog pairs seems to be ambiguous. I recommend that the authors elaborate how their finding is goldfish specific events or how similar with the other WGD events.

Minor comments

Page 13, para 2 "photoreceptor-specific genes are..."

The authors should check this sentence with attention to the following points; i) "patterns. photoreceptor-specific genes" might be "patterns, photoreceptor-specific genes", and ii) Is "gene expression" the opposite concept of the "amino acid mutations"? I also recommend that the authors site Figures in this paragraph.

Para 13 para 3 "These results suggest the possibility that wild goldfish acquired..."

The "wild goldfish" should be more specifically described. It seems that readers might confuse whether the wild goldfish means the ancestral goldfish before the domestication or the goldfish living under the natural condition.

Reviewer #2 (Remarks to the Author):

In this study, Kon et al. studied the evolution of ohnolog genes after goldfish-specific whole-genome duplication (WGD). Goldfish allotetraploidization provides a unique model to study early evolutionary events after WGD because it is one of the recent WGD events. Authors specifically study ohnolog gene expression at the single-cell level in the retina using RNA-seq and ATAC-seq, identified cell types, and the biases of L-ohnologs and S-ohnologs in retinal cell types. The manuscript is very well-written and easy to follow for the most part. The study also has important implications for genome evolution after WGD. However, some aspects should be clarified, and more analyses are needed which I detailed below.

Identification of ohnologs:

WGD/ohnologs identification includes two aspects: timing of duplication (i.e. all candidate ohnologs duplicated at the estimated time of WGD), and large conserved blocks of 1-to-2 synteny with the outgroup and with itself. Here, goldfish ohnologs are identified using zebrafish as an outgroup but all the goldfish paralogs that share the same zebrafish orthologs are included as ohnologs. This will inflate ohnolog numbers and include all the goldfish-specific small-scale duplicates (SSD) as ohnologs. Authors need to include synteny in goldfish ohnolog identification and remove potential SSDs from their analysis. Also, why are the ohnolog numbers higher than their previous report? Authors should systematically compare the source of newly identified ohnologs and their quality. Are there any recent SSD misclassified as ohnologs or are the changes due to genome improvement or something else? Ideally, I would expect that each gene in the ohnolog pair is represented only once because of the single genome duplication event. How many unique genes are there in 11,444 ohnolog pairs?

Single-cell RNA-seq:

Retina has multiple cell types of different shapes and sizes, and many of them can be problematic for scRNA-seq pipeline. Typically enzymatic digestion and 10x work the best for round cells. It is unclear to what extent the bias created by cell shapes and sizes impact the subsequent cell-type representations. Did the authors evaluate expected cell-type proportions independently? It should be tested and discussed in the manuscript because this can skew the results.

Cell types in the retina:

As a reader not an expert in retinal biology, it is confusing to me that the authors state in the introduction (lines 86-88) that there are 5 neuron and 3 glia cell types, but scRNA-seq identified 7 neurons and 5 non-neuronal cell types (lines 130-137). It will be useful to elaborate on what the new cell types or subtypes are. Also, the authors compared and show that cell types are similar in zebrafish and goldfish (lines 144-146). This is expected because zebrafish is used for cell-type annotation. A better approach is to independently evaluate if there are differences in cell types or their relative proportions in the two species.

Asymmetric subgenome expression:

I do not understand the point of Figure 2a. Is it not expected? What is the control here - randomly selected SSDs? The results in this section are expected as the authors showed in their previous papers because this is the sum of all cells and is similar to bulk RNA-seq.

Expression bias, dosage balance, and sub/neo-functionalization:

Are the differences presented in lines 188-195 statistically significant? It should be thoroughly evaluated using appropriate controls.

What is the rationale for using different criteria in identifying expression differences for dosage

balance and sub/neo-functionalizations? For example, lines 208-211 resulted in 611 ohnologs with different expressions between cell types, and lines 240-242 resulted in 632 pairs. It is unclear to me why the same criteria is not used for dosage balance and sub-neo-functionalization. Also, the authors state that "611 (5.3%) ohnolog pairs with no obvious difference in gene expression between the L-ohnolog and S-ohnolog in all cell types". However, asymmetric expression of L- and S-subgenome in different cell types is also due to dosage balance (see Lan and Pritchard 2016, Science).

It is unclear if the evaluation of sub/neo-functionalization and dosage balance is in comparison to zebrafish? Genes that are similarly expressed in zebrafish retinal cell types but different in goldfish L- or S-ohnologs will be the candidates for sub/neo-functionalization and dosage balance.

While single-cell analysis is very informative, for the most part, it validates the previous study on bulk tissues. It is crucial to show the unique aspects of ohnolog evolution that are identified in single cells but missed in bulk data. For example, examples of genes that are not sub/neo-functionalized in bulk data but they are in the single-cell. How many ohnologs are such, and so on.

ATAC-seq:

Lastly, this is perhaps the most underdeveloped part of the manuscript. Single-cell ATAC-seq is just used to validate certain aspects of scRNA-seq but the analysis falls short of identifying the regulatory regions that may underlie the observed patterns. Significant work is needed to integrate this with the rest of the manuscript and evaluate the potential regulatory regions that may explain the observed trends, rather than a small section to validate a few observations.

Overall, I find this study very interesting and potentially useful given it can be revised to include the above aspects.

Reviewer #3 (Remarks to the Author):

Kon et al present a single-cell mRNA and ATAC-seq dataset and analysis of the retinas from the goldfish *Carassius auratus*, and analyse patterns of evolution in gene duplicates following its recent whole genome duplication event (Cs4R), which resulted in two sub genomes (L and S). They confirm previously described global biases for expression from the L-subgenome, but further identify cell-type specific biases towards gene copies from both the L- and S-subgenomes. Functional enrichment for Gene Ontology categories for biased, and non-biased genes revealed some potentially interesting trends, and comparisons with data from zebrafish allowed the identification of potential cases of sub- and/or neo-functionalisation. Finally, the authors also present a sc-ATAC-seq dataset from the same tissue.

This is a very interesting dataset and quite useful for studying the evolution of gene expression after WGD and other types of gene duplication events, and I support the publication of the study. However, some changes to the interpretation of their results should be included, and I have included several suggestions below.

Comments:

1. The authors conclude that dosage compensation acts to maintain the expression of certain classes of genes (ribosomal, embryonic development, cell fate commitment) based on the observation that the L and S ohnologs do not show a significant difference in gene expression. Dosage compensation is typically viewed as maintaining a consistent level of total gene expression (combined levels of L- and S-ohnologs), rather than maintaining them at equal levels. In order to identify which genes are potentially being dosage compensated, the authors would need to identify the 'optimal' gene expression level, or the level pre-WGD. Compensation can be accomplished by halving the expression of both duplicates, eliminating the expression from one entirely, or something in between. The authors should correct the imprecise nature of their results.

2. I appreciate the usage of the zebrafish data to demonstrate neo/sub-functionalisation, however comparing relative levels of mRNA does not discount the possibility that lower expression levels

does not necessarily mean lower protein levels or loss of a gene's function in a particular cell type.

3. The authors include single-cell ATAC-seq data, but do little to compare this data to mRNA levels. I would like to see some sort of analysis comparing the genes that are L/S-biased (or not) by mRNA with those they identify as having differential accessibility by ATAC-seq - is there a significant overlap between these lists? Can the authors identify peaks specifically associated with expression biases (only accessible when the L or S ohnolog is more expressed)?

4. I think it would be useful to many readers who are interested in retinas and opsin genes to include some figures (supplemental perhaps) of opsin gene ohnolog expression patterns. Additionally, it seems a bit unclear why the authors chose the retina over another tissue.

5. The authors should provide more speculation about why genes might have evolved their observed biases between sub genomes. For example, are biased genes more likely to harbour amino acid substitutions? Additionally, is there an explanation for the global bias in gene expression from one sub genome?

6. At one point the authors state that there are 632 ohnolog pairs with cell type specific expression, but Figure 3a indicates that there are > 1000. I understand these differences are from pseudo-bulk vs cluster-based analyses, but it is quite confounding in the text. The authors should consider ways to clarify the different gene sets utilised.

7. The authors state that they use UMAP to compare the clustering based on L- or S-ohnologs only, however in the methods it is unclear if for these figures they also perform clustering and PCA using those same gene sets (or if they use the cluster labels from the initial clustering with all genes but with a different projection only). Sharing of scripts used for analysis would alleviate these issues. Additionally, it appears the authors used some sort of integration method to compare with zebrafish, but there are no details of this in the methods.

8. The authors could perform an enrichment test for whether or not S/L-biased genes are more likely to be cell-type specific (or not) than a random set of genes (Figure 3a).

9. Given the global bias towards expression from the L-subgenome, to me it would seem that equal expression of L- and S-ohnologs would indicate a potential over-expression of the S-copy. Can the authors somehow reconcile the global biases with the biases at the gene level?

The point-by-point answer to the referee's comments (Our comments were written in blue letters):

Reviewer #1:

The authors conducted an intensive transcriptome analysis in the goldfish retina. Their data of transcriptome analyses also carefully examined by using sophisticated statistic analysis. Based on these analyses, the authors provided the interpretation of how the expression levels of the duplicated genes have changed during evolutionary course. The data they collect would be useful for other researchers and should be published. However, I would like to make a few suggestions to the authors regarding the consistency of the introduction and the discussion as well as generalities of their findings. More specifically said, from this manuscript, it is unclear why the authors motivates for sticking to the combination of goldfish, retina and whole genome duplication. The authors should elaborate more on the biological or evolutionary significance of their focus. I indicated several difficult-to-understand points (Q1-Q4) in this manuscript (please see below).

Q1:

Page 4, para 1, “Therefore, determining the expression levels of ohnologues in a single cell...”
I assumed the authors aimed to narrow down their research focus from the first paragraph to second paragraph on Page 4. However, the contextual flow seems to be interrupted by the above indicated sentence.

As reviewer #1 suggested, we removed this sentence from the revised version of the manuscript (page 4, line 3; in the Introduction).

Q2:

Page 4, para 2 “Therefore, WGD is most likely to contributed to the evolution of retinal diversification in vertebrates, however, a detailed analysis on the process of gene evolution during the early stages after WGD remains unclear”

The context more diverged in the second paragraph on page 4. The authors raised different issues in this paragraph. They explained the specific feature of the retina of vertebrates as a light-sensitive tissue and the two round of WGD which occurred in the common ancestor of the vertebrates. My understanding is that the authors want to investigate how recently duplicated gene

evolved at the level of gene expression patterns by using retina; I hope my understanding is correct. However, to me, it was unclear why did the authors had to use the retina, rather than the other tissues. The retina is not the sole diversified tissues in the lineage of the vertebrate relating with WGD. Sophisticate skeletons, brains, and etc. are also known as the diversified tissue and organs in the vertebrate lineages. I recommend that the author more carefully explain their motivation why they focus on the retina.

As reviewer #1 suggested, we added the reason for using the retina for scRNA-seq analysis.

(page 4, line 14; in the Introduction)

“In this study, we focused on single-cell transcriptome analysis of the retina for the following three reasons. First, in the previous study, the retina and the brain showed the most diversified expression between ohnologs after the recent WGD²³. The retina has a wide variety of cell types expressing cell type specific genes with simpler tissue structure³⁵. Second, to date, single-cell RNA-seq (scRNA-seq) analysis has been used to study the retina of various vertebrate species, including zebrafish, chicken, mice, and primates^{26,35-40}. These studies provide datasets for the cell-type specificity of thousands of genes and demonstrate that the retina is an ideal tissue for studying the cell specificity of gene expression in vertebrates. Third, vertebrate retinas have highly diversified functions despite maintaining common basic cellular components and tissue structure, better allowing to compare the expression patterns between species.”

Q3 (continued from the above)

Discussion

Page 12 para 4 “This is consistent with findings that...”

Page 13, para 1 “The evolutionary diversity associate with visual functions...”

The authors explained their finding about the analyses. On the line 341, they described that their results are consistent with previous studies on plants and vertebrates, but the next paragraph is simply explained the visual functions of vertebrates. Although I know that the authors use the retina for their research, it is hard for me to understand how the former and the latter paragraphs are connected.

I strongly recommend that the authors consider this discussion part, taking into consideration the above comment (Q2).

As reviewer #1 suggested, we explained the connection among those paragraphs as follows. We

tried to better clarify our intended meaning from this addition and our additional explanation in Q2.

(page 14, line 7; in Discussion)

“Ohnolog pairs with sub/neo-functionalization are often associated with the diversified tissues among species over evolution.”

Q4.

Page 13, para 2 “We identified 306 ohnolog pairs...”

From this paragraph, the finding of the diversified expression patterns is described. However, from these descriptions, it is hard to distinguish their finding is the goldfish specific evolutionary events or comparable with the other WGD events. Although the statements about four genes (cabp5a, cabp5b, rs1a, and ptgs2b) are relatively specific in the next paragraph (line 363), the statement about 306 ohnolog pairs seems to be ambiguous. I recommend that the authors elaborate how their finding is goldfish specific events or how similar with the other WGD events.

Minor comments

To distinguish between goldfish-specific WGD events and the other WGD events, we named goldfish-specific WGD events as “Cs4R” in the revised manuscript (page 13, line 23, 26; in Discussion).

We agree that it is very interesting to compare other WGD events, as reviewer #3 mentioned. However, we were unable to do it at this time as no retinal scRNA-seq data from *Xenopus* or common carp is available. In the future, we want to study this issue in depth.

It is known that most rod and cone phototransduction-related genes (e.g., *CNGA1* vs *CNGA3*) were generated on 1R/2R WGD (Lamb, et al., 2016). Accordingly, we did not observe rod and cone phototransduction-related genes in these 306 ohnolog pairs (table S13). Therefore, additional sub/neo-functionalization of rod and cone phototransduction-related genes after Cs4R seems unlikely.

We added the following text in the revised manuscript as follows:

(page 14, line 26; in Discussion)

“It is known that most rod and cone phototransduction-related genes (e.g., *CNGA1* vs *CNGA3*) were generated on 1R/2R WGD (Lamb, et al., 2016). Therefore, additional sub/neo-functionalization of rod and cone phototransduction-related genes after Cs4R seems unlikely.”

Page 13, para 2 “photoreceptor-specific genes are...”

The authors should check this sentence with attention to the following points; i) “patterns. photoreceptor-specific genes” might be “patterns, photoreceptor-specific genes and ii) Is “gene expression” the opposite concept of the “amino acid mutations”? I also recommend that the authors site Figures in this paragraph.

Thank you for pointing this issue out. We replaced the text as follows:

(page 14, line 23; in Discussion)

“This finding suggests that the evolution of these phototransduction ohnologs is unlikely to deviate from photoreceptor-specific expression patterns, and that photoreceptor-specific genes are more likely to evolve through other mechanisms such as post-transcriptional modification or amino acid mutation.”

Para 13 para 3 “These results suggest the possibility that wild goldfish acquired...”

The “wild goldfish” should be more specifically described. It seems that readers might confuse whether the wild goldfish means the ancestral goldfish before the domestication or the goldfish living under the natural condition.

To address reviewer #1’s concern, we changed “wild goldfish” to “goldfish ancestor” in the revised manuscript (page 15, line 11; in the Discussion).

Reviewer #2 (Remarks to the Author):

In this study, Kon et al. studied the evolution of ohnolog genes after goldfish-specific whole-genome duplication (WGD). Goldfish allotetraploidization provides a unique model to study early evolutionary events after WGD because it is one of the recent WGD events. Authors specifically study ohnolog gene expression at the single-cell level in the retina using RNA-seq and ATAC-seq, identified cell types, and the biases of L-ohnologs and S-ohnologs in retinal cell types. The manuscript is very well-written and easy to follow for the most part. The study also has important implications for genome evolution after WGD. However, some aspects should be clarified, and more analyses are needed which I detailed below.

1) Identification of ohnologs:

WGD/ohnologs identification includes two aspects: timing of duplication (i.e., all candidate ohnologs duplicated at the estimated time of WGD), and large conserved blocks of 1-to-2 synteny with the outgroup and with itself. Here, goldfish ohnologs are identified using zebrafish as an outgroup but all the goldfish paralogs that share the same zebrafish orthologs are included as ohnologs. This will inflate ohnolog numbers and include all the goldfish-specific small-scale duplicates (SSD) as ohnologs. Authors need to include synteny in goldfish ohnolog identification and remove potential SSDs from their analysis. Also, why are the ohnolog numbers higher than their previous report? Authors should systematically compare the source of newly identified ohnologs and their quality. Are there any recent SSD misclassified as ohnologs or are the changes due to genome improvement or something else? Ideally, I would expect that each gene in the ohnolog pair is represented only once because of the single genome duplication event. How many unique genes are there in 11,444 ohnolog pairs?

We think our insufficient explanation may have misled reviewer #2. To increase clarity, we added a figure describing the detailed pipeline for selecting the 11,444 pairs of ohnologs (Supplemental Figure S1a, right panel). As we described in the Materials and Methods and this figure, we used a combination of the reported recently genome assembly generated by Hi-C scaffolding (Chen et al., 2020) and the NCBI RefSeq annotation for the goldfish genome. The new genome assembly is more continuous and contains fewer unplaced contigs. The NCBI RefSeq annotation is generated by the NCBI Gnomon annotation pipeline. This pipeline uses the latest RNA-seq reads and ortholog sequences in the NCBI database. NCBI RefSeq annotations showed longer CDS sequences than our previous annotations (Reviewer Fig. 1). These factors contributed to better identification of ohnolog pairs; thus, we successfully obtained a higher number of ohnolog pairs

(11,444 pairs) than a previous study (Chen et al., 2019).

Reviewer Figure 1. CDS length distribution of the two different genome annotations

We chose only pairs mapped on homeologous chromosome pairs; thus, our goldfish ohnolog identification strategy includes synteny information. In this strategy, most goldfish-specific small-scale duplicates (SSD) were excluded because they were expected to be localized to nonhomeologous chromosomes. To confirm the synteny localization of ohnolog pairs on homeologous chromosomes, we mapped ohnolog pairs on the goldfish genome and found that most ohnolog pairs were mapped on syntenic regions of homeologous chromosomes (Figure S1d-f). Further, we tested whether the 11,444 ohnolog pairs formed colinear synteny blocks, finding that 97.1% of the ohnolog pairs formed statistically significant colinear synteny blocks. We modified the text in the Results and Materials and Methods as follows:

(page 5, line 8; in the Results)

“We newly defined high-quality ohnolog pairs in goldfish to analyze preservation and divergence of L-ohnolog and S-ohnolog gene expressions using the scRNA-seq dataset (Fig. S1a, right panel).”

(page 5, line 14; in the Results)

“Thus, these 11,444 ohnolog pairs in goldfish have 11,444 corresponding zebrafish orthologs, located on their orthologous chromosomes (Fig. S1d).”

(page 17, line 19; in the Materials and Methods)

“Using MCSanX, we tested whether the 11,444 ohnolog pairs formed colinear synteny blocks, finding that 97.1% of the ohnolog pairs formed statistically significant colinear synteny blocks (Fig. S1d-f).”

(page 17, line 20; in the Materials and Methods)

“Most goldfish specific small-scale duplicates (SSD) were excluded in this strategy as they are expected to be localized to nonhomeologous chromosomes.”

We added Supplemental Figure S1a right panel and S1d-f to the revised manuscript.

(Supplemental Figure S1 Legends)

“Right panel; Pipeline for identification of 11,444 ohnolog pairs in goldfish genome.”

“(d) Oxford dot plot of the 11,444 ohnolog pairs. (e) Dual synteny plot of the 11,444 ohnolog pairs. (f) Bar plot showing the chromosomal position of each ohnolog.”

2) Single-cell RNA-seq:

Retina has multiple cell types of different shapes and sizes, and many of them can be problematic for scRNA-seq pipeline. Typically enzymatic digestion and 10x work the best for round cells. It is unclear to what extent the bias created by cell shapes and sizes impact the subsequent cell-type representations. Did the authors evaluate expected cell-type proportions independently? It should be tested and discussed in the manuscript because this can skew the results.

To confirm whether the data of cell composition in the scRNA-seq analysis of goldfish reflects tissue cell composition, we measured cell-type ratios by a different method. The vertebrate retina has three nuclear layers and two plexiform layers where synaptic connections are formed. The outer nuclear layer (ONL) is composed of the photoreceptor bodies, and the inner nuclear layer (INL) is composed of the cell bodies of bipolar, horizontal, and amacrine cells. Meanwhile, the ganglion cell layer (GCL) is mainly composed of ganglion cell bodies. In the vertebrate retina, each cell-type forms a layered structure, and the number of cells in each layer can be counted to determine the approximate number of cell types. Accordingly, we prepared frozen sections from Wakin retinal tissues, stained with DAPI to stain the nuclei of all retinal cells and counted the number of cells in the different layers of the retina. The results showed $33.1 \pm 6.0\%$ of ONL, $53.6 \pm 7.8\%$ of inner nuclear layer (INL), and $13.2 \pm 2.1\%$ of GCL. These values are roughly consistent with the result of our scRNA-seq analysis (29.7% ONL, 54.7% INL, and 1.5% GCL). The number of cells in the GCL in scRNA-seq seems to be smaller than expected from the frozen section experiment. This can be explained because the GCL also contains some amacrine cells. These results are shown in Supplemental Figure S3c, d, and we added the following text:

(page 6, line 10 in the Results)

“To confirm whether the composition of each cell-type in the scRNA-seq analysis of goldfish reflects tissue cell composition, we measured cell-type ratios by a different method. The vertebrate retina has three nuclear layers and two plexiform layers where synaptic connections are formed³⁵. The outer nuclear layer (ONL) is composed of the photoreceptor bodies, and the inner nuclear layer (INL) is composed of the cell bodies of bipolar, horizontal, and amacrine cells. Meanwhile, the ganglion cell layer (GCL) is mainly composed of ganglion cell bodies. In the vertebrate retina, each cell-type forms a layered structure, and the number of cells in each layer can be counted to determine the approximate number of cell types. Accordingly, we prepared frozen sections from Wakin retinal tissues, stained with DAPI to stain the nuclei of all retinal cells and counted the number of cells in the different layers of the retina (Fig. S3c). The results showed $33.1 \pm 6.0\%$ of ONL, $53.6 \pm 7.8\%$ of INL, and $13.2 \pm 2.1\%$ of GCL. These values are roughly consistent with the result of our scRNA-seq analysis (29.7% ONL, 54.7% INL, and 1.5% GCL). The number of cells in the GCL in scRNA-seq seems to be smaller than expected from the frozen section experiment. This can be explained because the GCL also contains some amacrine cells.

(page 18, line 19; in the Materials and Methods)

“DAPI staining of the Wakin retinal section

We prepared frozen sections from Wakin retinal tissue, performed DAPI staining as previously described (Kon et al 2020), and counted the number of cells in different layers of the retina.”

(Supplemental Figure S3 Legends)

“(c) DAPI staining of goldfish retinal sections. The outer nuclear layer (ONL), the inner nuclear layer (INL), and the ganglion cell layer (GCL) are shown.”

3) Cell types in the retina:

As a reader not an expert in retinal biology, it is confusing to me that the authors state in the introduction (lines 86-88) that there are 5 neuron and 3 glia cell types, but scRNA-seq identified 7 neurons and 5 non-neuronal cell types (lines 130-137). It will be useful to elaborate on what the new cell types or subtypes are. Also, the authors compared and show that cell types are similar in zebrafish and goldfish (lines 144-146). This is expected because zebrafish is used for cell-type annotation. A better approach is to independently evaluate if there are differences in cell types or their relative proportions in the two species.

As reviewer #2 pointed out, the numbers of cell types in the Introduction and Results are different. In the Results, we included all cell subtypes, which is the reason for the difference. We

modified the text to explain the rationale for the difference.

(page 4, line 6 in the Introduction)

“The vertebrate retina is a light-sensitive tissue consisting of seven major classes of neurons (rod/cone photoreceptor cells, horizontal cells, bipolar cells, ganglion cells, and GABAergic/glycinergic amacrine cells) and three classes of glia (Müller glia, microglia, and oligodendrocytes) accompanied by retinal pigmented epithelial cells and vascular endothelial cells”

In addition, we compared the cell composition of the zebrafish and goldfish retinas (new Supplemental Fig S3d). The numbers in goldfish and zebrafish are roughly similar, however, the most significant difference is the 25.7% rod photoreceptor cells in goldfish compared to only 1.0% in zebrafish. Previous histological studies suggest that the ratio of photoreceptor cells in the retina should be similar between these two species. However, in zebrafish, it is much lower than in tissue sections. Since photoreceptors have long outer segments and connecting cilia, they may be more susceptible to differences in experimental conditions. Differences in filtering and annotating cells during data analysis may also contribute to the differences in the number of rod cells (Hoang et al., 2020). No major differences were noted among cones, bipolar, and amacrine, suggesting that there were no critical issues in the scRNA-seq analysis performed in this study. These results are described as follows:

(page 6, line 23; in the Results)

“We compared the cell composition of the zebrafish and goldfish retinas (Supplemental Fig S3d). The numbers in goldfish and zebrafish are roughly similar, however, the most significant difference is the 25.7% rod photoreceptor cells in goldfish compared to only 1.0% in zebrafish. Previous histological studies suggest that the ratio of photoreceptor cells in the retina should be similar between these two species. However, in zebrafish, it is much lower than in tissue sections. Since photoreceptors have long outer segments and connecting cilia, they may be more susceptible to differences in experimental conditions. Differences in filtering and annotating cells during data analysis may also contribute to the differences in the number of rod cells (Hoang et al., 2020). No major differences were noted among cones, bipolar, and amacrine, suggesting that there were no critical issues in the scRNA-seq analysis performed in this study.”

(Supplemental Figure S3 Legends)

“(d) Percentage of cell types identified by scRNA-seq analysis in goldfish and zebrafish. Twelve cell types identified were shown. Abbreviations: V/E, vascular endothelial cells; RPE, retinal

pigmented epithelial cells; MG, Müller glia; RGC, retinal ganglion cells; HC, horizontal cells; AC amacrine cells; BC, bipolar cells.”

4) Asymmetric subgenome expression:

I do not understand the point of Figure 2a. Is it not expected? What is the control here - randomly selected SSDs? The results in this section are expected as the authors showed in their previous papers because this is the sum of all cells and is similar to bulk RNA-seq.

As reviewer #2 pointed out, the result in Fig2a is expected from previous bulk RNA-seq data. The red line indicates a simple $y = x$ line with equal expression levels of S- and L-ohnologs. The slope of the linear regression is significantly < 1 indicating that the total expression level of all L-ohnologs in each cell is systematically higher than that of all S-ohnologs. This result is related to the subsequent Figs 2b, 2c, and 2d. Fig 2a represents a positive control, which indicates the integrity of our single-cell data sets. As we believe that this is relevant positive data, we did not remove it from Figure 2. However, if reviewer #2 considers it better, we can move it to the supplemental figures.

5) Expression bias, dosage balance, and sub/neo-functionalization:

Are the differences presented in lines 188-195 statistically significant? It should be thoroughly evaluated using appropriate controls.

What is the rationale for using different criteria in identifying expression differences for dosage balance and sub/neo-functionalizations? For example, lines 208-211 resulted in 611 ohnologs with different expressions between cell types, and lines 240-242 resulted in 632 pairs. It is unclear to me why the same criteria is not used for dosage balance and sub-neo-functionalization. Also, the authors state that "611 (5.3%) ohnolog pairs with no obvious difference in gene expression between the L-ohnolog and S-ohnolog in all cell types." However, asymmetric expression of L- and S-subgenome in different cell types is also due to dosage balance (see Lan and Pritchard 2016, Science).

As requested, we tested whether S/L-biased genes show more cell-type specific expression than a random set of genes using Fisher's exact test and found that there were no significant difference between them. We added the following text:

(page 8, line 18 in the Results)

“We performed an enrichment test to determine whether L/S-biased genes show more cell-type specific expression than a random set of genes. The result showed no significant difference between them (Fisher's exact test, $p = 0.093$).”

As reviewer #2 pointed out, our description might have been confusing (similar to comment 6 from reviewer #3). We chose 611 ohnolog pairs “with unbiased expression,” these are not ohnolog pairs with different expressions between cell types. We chose 1070 ohnolog pairs (page 6, line 9) based on cell-type specificity of the total expression of L- and S-ohnologs to find overall cell type-specific genes; in contrast, we selected the 632 ohnolog pairs based on simple individual expression of L- or S-ohnologs to identify sub/neo-functionalized ohnolog pairs. To clarify how we selected ohnolog pairs, we added the following text to the Results:

(page 8, line 11 in the Results)

“We found that 1,070 ohnolog pairs (9.3%) showed cell type specific expression profiles (cell-type specificity of the total expression of L- and S-ohnologs)”

(page 9, line 31 in the Results)

“First, we selected the cell type specific ohnologs among the 11,444 ohnolog pairs by searching for ohnologs with significantly higher expression in a particular cell-type (cell-type specificity of individual expression of L- or S ohnolog).”

We think that the discussion regarding “dosage balance” is similar to our reply to comment 1 of reviewer #3. Equal expression levels of ohnologs are essential for “dosage balanced” pairs. However, the total amount of ohnolog pairs is not relevant. In most cases, they will be twice those of ohnolog’s ancestor genes. Hox gene clusters are a good example for this. However, since our description may be misleading, we decided to change “dosage balance expression” to “unbiased expression” in the revised manuscript.

It is unclear if the evaluation of sub/neo-functionalization and dosage balance is in comparison to zebrafish? Genes that are similarly expressed in zebrafish retinal cell types but different in goldfish L- or S-ohnologs will be the candidates for sub/neo-functionalization and dosage balance.

While single-cell analysis is very informative, for the most part, it validates the previous study on bulk tissues. It is crucial to show the unique aspects of ohnolog evolution that are identified in single cells but missed in bulk data. For example, examples of genes that are not sub/neo-functionalized in bulk data but they are in the single-cell. How many ohnologs are such, and so

on.

In this analysis, we did not evaluate sub/neo-functionalization and dosage balance in comparison to zebrafish. Since the analytical platform differs between the zebrafish study and ours, it is technically difficult to compare them. Accordingly, to clarify these differences we added the expression patterns in zebrafish in the new Figure S7. The gene expression patterns in zebrafish were roughly similar to those of L- or S-ohnologs.

(Supplemental Figure S7 Legends)

“The expression patterns of zebrafish orthologs are also shown in green circles.”

We tested whether the sub/neo-gene cluster found on scRNA-seq data (306 ohnolog pairs) overlapped with a previously reported sub/neo-gene cluster found in bulk RNA-seq of tissue (368 ohnolog pairs; Chen et al., 2019). This analysis showed that only four genes (*oxr1b*, *rgs16*, *nat16*, and *tubb2*) overlapped in both groups, indicating that the sub-value genes found by scRNA-seq are significantly different from those obtained by bulk RNA-seq analysis. This indicates that scRNA-seq analysis is useful for identifying sub/neo-evolved genes.

(page 10, line 15; in Results)

“We tested whether the sub/neo-functionalized gene cluster found on scRNA-seq data (306 ohnolog pairs) overlapped with a previously reported sub/neo-gene cluster found in bulk RNA-seq of tissue (368 ohnolog pairs¹¹). This analysis showed that only four genes (*oxr1b*, *rgs16*, *nat16*, and *tubb2*) overlapped in both groups, indicating that the sub-value genes found by scRNA-seq are significantly different from those obtained by bulk RNA-seq analysis. This indicates that scRNA-seq analysis is useful for identifying sub/neo-evolved genes.”

6) ATAC-seq:

Lastly, this is perhaps the most underdeveloped part of the manuscript. Single-cell ATAC-seq is just used to validate certain aspects of scRNA-seq but the analysis falls short of identifying the regulatory regions that may underlie the observed patterns. Significant work is needed to integrate this with the rest of the manuscript and evaluate the potential regulatory regions that may explain the observed trends, rather than a small section to validate a few observations.

This comment is similar to comment 3 of reviewer #3. As both reviewers suggested, we

integrated the potential regulatory regions obtained from scATAC-seq into the selected data sets from scRNA-seq data. We also tried to find the DNA binding sequence of retinal transcription factors such as Otx2 and Crx. In other studies using human and mice genome, it was easier to find DNA binding sequences in open chromatin regions because their genome senesces were well refined and the analysis pipelines are well established. In our study, we used a goldfish genome sequence, of possibly lower quality than that for typical model organisms such as humans and mice. Therefore, we prepared a pipeline to find DNA binding sequences in open chromatin regions (OCR); however, we only found significant enrichment of Otx2 DNA binding sequences in rods, cones, and BC clusters. These data are illustrated in Figures 6d and 7. We believe that these additional data significantly improved the manuscript.

(page 11, line 25 in the Results)

“Then, we focused on the potential regulatory regions underlying the expression patterns found on scRNA-seq analysis. We compared significant peaks of OCRs in the scATAC-seq data with mRNA expression patterns on scRNA-seq data. To evaluate the relationship between these two methods, we calculated a Pearson correlation coefficient matrix between the observed cell types for each gene set. We found significantly high correlations between the corresponding cell types for any gene set in scRNA-seq and scATAC-seq (Fig. 6d), suggesting that gene the expression profiles in retinal cell types from our scRNA-seq analysis and the OCR profiles in retinal cell types scATAC-seq data have essentially similar features.

Using these data, we predicted the candidate regulatory regions of the bias-expressed genes (L-bias 2,690 ohnolog pairs, 6,083 OCRs, Table S15; S-bias 2,443 ohnolog pairs, 5,810 OCRs, Table S16) and sub/neo-functionalization genes (306 ohnolog pairs, 849 OCRs, Table S17) based on scATAC-seq analysis. These data provide a comprehensive view of the evolution of potential gene regulatory regions after whole genome duplication of the goldfish ancestor genome.”

(page 12, line 14 in the Results)

“Next, we focused on the evolution of regulatory regions for cell-type-specific ohnolog pairs. We identified OCRs for sub/neo-functionalization genes (306 ohnolog pairs, 849 OCRs, Table S17). These data sets are therefore candidates for cell type-specific, subgenome-evolved regulatory regions. We further used this data set to search for Otx2/Crx binding sequences (Fig. 7a) with photoreceptor/bipolar cell-specific expression in the retina. This analysis revealed that Otx2/Crx binding sequences are significantly enriched in the OCRs of these cell types ($P < 1.0e-15$, Wilcoxon test, Fig. 7b). We described 89 OCRs containing Otx2/Crx binding sites for regulation of ohnolog pairs with photoreceptor/bipolar cell-specific expression (Table S18). These OCRs are candidate regulatory regions for cell type specificity of differentially expressed

ohnolog pairs between L/S-subgenomes. For example, scRNA-seq analysis showed that prph2b, a photoreceptor specific glycoprotein, is more highly expressed in the L-subgenome than in the S-subgenome (Fig. 7c). In addition, scATAC-seq analysis revealed significant OCRs with Otx2/Crx binding sites in the L-subgenome but not in the S-subgenome. Accordingly, our scATAC-seq data provide information on gene regulatory regions important for the evolution of gene expression in retinal ohnolog pairs.”

(page 18, line 1 in the Materials and Methods)

“We searched for the Otx2/Crx binding motif (TAATCC) in the genome sequence and defined the regions extending from 100 bp upstream to 100 bp downstream of Otx2/Crx binding sequences. For each cell-type, we compared the average expression value in scRNA-seq and the average scATAC-seq signals for each gene.”

(page 36, line 5 in Figure 7’s Legend)

“Figure 7. Analysis of Otx2/Crx binding sequence based on scATAC-seq.

(a) Binding motif of Crx and Otx2. IDs in parentheses come from JASPAR. (b) Median value of normalized scATAC-seq signals around the Crx/Otx2 binding motif in each OCR are higher in rods, cones, and BC than in other cell types. (c) Expression pattern of the prph2b ohnolog pair (example of L-subgenome dominant ohnolog pair) based on scRNA-seq analysis data (upper panel). OCR peaks of the prph2b ohnolog pair are based on scATAC-seq data (lower panel). The expression data from scRNA-seq analysis is consistent with the results of scATAC-seq analysis.”

(Supplemental Table Titles)

“Table S15. Candidate regulatory regions of bias-expressed genes (L-bias 2,690 ohnolog pairs, 6083 OCRs) based on scATAC-seq analysis.

Table S16. Candidate regulatory regions of bias-expressed genes (S-bias 2,443 ohnolog pairs, 5810 OCRs) based on scATAC-seq analysis.

Table S17. Candidate regulatory regions of sub/neo-functionalization genes (306 ohnolog pairs, 849 OCRs) based on scATAC-seq analysis.

Table S18. OCRs containing Otx2/Crx binding sites for regulation of the ohnolog pairs with photoreceptor/bipolar cell-specific expression.”

Reviewer #3 (Remarks to the Author):

Kon et al present a single-cell mRNA and ATAC-seq dataset and analysis of the retinas from the goldfish *Carassius auratus*, and analyse patterns of evolution in gene duplicates following its recent whole genome duplication event (Cs4R), which resulted in two sub genomes (L and S). They confirm previously described global biases for expression from the L-subgenome, but further identify cell-type specific biases towards gene copies from both the L- and S-subgenomes. Functional enrichment for Gene Ontology categories for biased, and non-biased genes revealed some potentially interesting trends, and comparisons with data from zebrafish allowed the identification of potential cases of sub- and/or neo-functionalisation. Finally, the authors also present a sc-ATAC-seq dataset from the same tissue.

This is a very interesting dataset and quite useful for studying the evolution of gene expression after WGD and other types of gene duplication events, and I support the publication of the study. However, some changes to the interpretation of their results should be included, and I have included several suggestions below.

Comments:

Q1

1. The authors conclude that dosage compensation acts to maintain the expression of certain classes of genes (ribosomal, embryonic development, cell fate commitment) based on the observation that the L and S ohnologs do not show a significant difference in gene expression. Dosage compensation is typically viewed as maintaining a consistent level of total gene expression (combined levels of L- and S-ohnologs), rather than maintaining them at equal levels. In order to identify which genes are potentially being dosage compensated, the authors would need to identify the ‘optimal’ gene expression level, or the level pre-WGD. Compensation can be accomplished by halving the expression of both duplicates, eliminating the expression from one entirely, or something in between. The authors should correct the imprecise of their results.

As reviewer #3 mentioned, “Dosage compensation” is usually used for neutralization counteraction of unbalanced gene expression due to unequal distribution of sex chromosomes (Marin et al., 2000 PMID: 11084626). In this case, as referee #3 mentions, the “total amount” of the genes in the cells expressed from two ohnologs should be maintained. We previously showed “dosage balance” genes (Glasauer et al., 2014, Makino et al., 2010 PMID: 20439718) and the essential role of the equal expression levels of ohnologs. However, the total amount of ohnolog

pairs is not important. In most cases, they will be twice those of ohnolog's ancestor genes. Hox gene clusters are a good example of this situation. However, as our description may be misleading, we decided to change “dosage balance expression” to “unbiased expression” in the revised manuscript.

Q2

2. I appreciate the usage of the zebrafish data to demonstrate neo/sub-functionalisation, however comparing relative levels of mRNA does not discount the possibility that lower expression levels does not necessarily mean lower protein levels or loss of a gene's function in a particular cell type.

As reviewer #3 mentioned, it is difficult to completely rule out that reduced gene expression does not result in reduced protein levels or loss of gene function in a particular cell-type. However, as ohnolog pairs show highly similar sequences, it is likely that their protein stability and/or activity is similar in the cells. Thus, in most cases, the mRNA expression levels of ohnolog pairs will most likely reflect protein amount and/or activity. We added the following text to the Discussion:

(page 14, line 28, in Discussion)

“Although this study focused primarily on the expression levels of mRNA ohnolog pairs, it is difficult to completely rule out that reduced gene expression does not result in reduced protein levels or loss of gene function in a particular cell-type. However, because ohnolog pairs show highly similar sequences, it is likely that their protein stability and/or activity are similar in the cells. Thus, in most cases, the mRNA expression levels of ohnolog pairs will most likely reflect protein amount and/or activity.”

Q3 ATAC

3. The authors include single-cell ATAC-seq data, but do little to compare this data to mRNA levels. I would like to see some sort of analysis comparing the genes that are L/S-biased (or not) by mRNA with those they identify as having differential accessibility by ATAC-seq - is there a significant overlap between these lists? Can the authors identify peaks specifically associated with expression biases (only accessible when the L or S ohnolog is more expressed)?

This comment is similar to comment 6 of reviewer #2. We observed similar L/S-bias from scATAC-seq data to that from scRNA-seq data (Fig S6c).

As reviewers suggested, we integrated the potential regulatory regions obtained from scATAC-

seq into the selected data sets from scRNA-seq data. We also tried to find the DNA binding sequence of retinal transcription factors such as Otx2 and Crx. In other studies using human and mice genome, it was easier to find DNA binding sequences in OCR because their genome sequences were well refined and the analysis pipelines are well established. In our study, we used a goldfish genome sequence, of possibly lower quality than that for typical model organisms such as humans and mice. Therefore, we prepared a pipeline to find DNA binding sequences in OCR; however, we only found significant enrichment of Otx2 DNA binding sequences in rods, cones, and BC clusters. These data are illustrated in Figures 6d and 7. We believe that these additional data significantly improved the manuscript.

(page 11, line 25 in the Results)

“Then, we focused on the potential regulatory regions underlying the expression patterns found on scRNA-seq analysis. We compared OCRs in the scATAC-seq data with mRNA expression patterns on scRNA-seq data. To evaluate the relationship between these two methods, we calculated a Pearson correlation coefficient matrix between the observed cell types for each gene set. We found significantly high correlations between the corresponding cell types for any gene set in scRNA-seq and scATAC-seq (Fig. 6d), suggesting that gene the expression profiles in retinal cell types from our scRNA-seq analysis and the OCR profiles in retinal cell types scATAC-seq data have essentially similar features.

Using these data, we predicted the candidate regulatory regions of the bias-expressed genes (L-bias 2,690 ohnolog pairs, 6083 OCRs, Table S15; S-bias 2443 ohnolog pairs, 5810 OCRs, Table S16) and sub/neo-functionalization genes (306 ohnolog pairs, 849 OCRs, Table S17) based on scATAC-seq analysis. These data provide a comprehensive view of the evolution of potential gene regulatory regions after whole genome duplication of the goldfish ancestor genome.”

(page 12, line 14 in the Results)

“Next, we focused on the evolution of regulatory regions for cell-type-specific ohnolog pairs. We identified OCRs for sub/neo-functionalization genes (306 ohnolog pairs, 849 OCRs, Table S17). These data sets are therefore candidates for cell type-specific, subgenome-evolved regulatory regions. We further used this data set to search for Otx2/Crx binding sequences (Fig. 7a) with photoreceptor/bipolar cell-specific expression in the retina. This analysis revealed that Otx2/Crx binding sequences are significantly enriched in the OCRs of these cell types ($P < 1.0e-15$, Wilcoxon test, Fig. 7b). We described 89 OCRs containing Otx2/Crx binding sites for regulation of ohnolog pairs with photoreceptor/bipolar cell-specific expression (Table S18). These OCRs are candidate regulatory regions for cell type specificity of differentially expressed ohnolog pairs between L/S-subgenomes. For example, scRNA-seq analysis showed that prph2b,

a photoreceptor specific glycoprotein, is more highly expressed in the L-subgenome than in the S-subgenome (Fig. 7c). In addition, scATAC-seq analysis revealed significant OCRs with Otx2/Crx binding sites in the L-subgenome but not in the S-subgenome. Accordingly, our scATAC-seq data provide information on gene regulatory regions important for the evolution of gene expression in retinal ohnolog pairs.”

(page 18, line 1 in the Materials and Methods)

“We searched for the Otx2/Crx binding motif (TAATCC) in the genome sequence and defined the regions extending from 100 bp upstream to 100 bp downstream of Otx2/Crx binding sequences. For each cell-type, we compared the average expression value of each gene in scRNA-seq and the average scATAC-seq OCR signals related to the corresponding gene.”

(page 35, line 6 in Figure 6’s Legend)

“(d) Pearson correlation coefficient matrix between observed cell types for each of the gene sets using the data from scRNA-seq and scATAC-seq analysis. The correlations between scRNA-seq and scATAC-seq are significantly high between the corresponding cell types for any gene set (left panel, 2,690 ohnolog pairs showed biased expression toward the L-ohnolog over the S-ohnolog; middle panel, 2,433 ohnolog pairs showed biased expression toward the S-ohnolog over the L-ohnolog; right panel, 306 ohnolog pairs with sub/neo-functionalization).”

(page 36, line 5 in Figure 7’s Legend)

“Figure 7. Analysis of Otx2/Crx binding sequence based on scATAC-seq.

(a) Binding motif of Crx and Otx2. IDs in parentheses come from JASPAR. (b) Median value of normalized scATAC-seq signals around the Crx/Otx2 binding motif in each OCR are higher in rods, cones, and BC than in other cell types. (c) Expression pattern of the prph2b ohnolog pair (example of L-subgenome dominant ohnolog pair) based on scRNA-seq analysis data (upper panel). OCR peaks of the prph2b ohnolog pair are based on scATAC-seq data (lower panel). The expression data from scRNA-seq analysis is consistent with the results of scATAC-seq analysis.”

Q4 opsin

4. I think it would be of useful to many readers who are interested in retinas and opsin genes to include some figures (supplemental perhaps) of opsin gene ohnolog expression patterns. Additionally, it seems a bit unclear why the authors chose the retina over another tissue.

As reviewer #3 suggested, we presented the opsin expression pattern in Supplemental Figure S9a

and added the corresponding commentary to the revised text.

(page 10, line 27 in the Results)

“Consistent with this, rhodopsin (*rho*) and *opn1lw1* showed non-divergent expression in rods and cones, respectively (Fig. S9a). Despite the low expression levels, both L/S-ohnologs of *opn1mw4* and *opn1sw2* are expressed in cones.”

(Supplemental Figure S9 Legends)

“(a) Expression of opsin L/S ohnolog pairs. The gene expression levels of opsin ohnologs in the goldfish retina are shown.”

The reason for choosing the retina for this scRNA-seq analysis (the same question as reviewer #1 comment 2) was added as follows:

(page 4, line 14; in the Introduction)

“In this study, we focused on single-cell transcriptome analysis of the retina for the following three reasons. First, in the previous study, the retina and the brain showed the most diversified expression between ohnologs after the recent WGD²³. The retina has a wide variety of cell types expressing cell type specific genes with simpler tissue structure³⁵. Second, to date, single-cell RNA-seq (scRNA-seq) analysis has been used to study the retina of various vertebrate species, including zebrafish, chicken, mice, and primates^{26,35–40}. These studies provide datasets for the cell-type specificity of thousands of genes and demonstrate that the retina is an ideal tissue for studying the cell specificity of gene expression in vertebrates. Third, vertebrate retinas have highly diversified functions despite maintaining common basic cellular components and tissue structure, better allowing to compare the expression patterns between species”

Q5.

5. The authors should provide more speculation about why genes might have evolved their observed biases between sub genomes. For example, are biased genes more likely to harbour amino acid substitutions? Additionally, is there an explanation for the global bias in gene expression from one sub genome?

As reviewer #3 suggested, we added the following text to the Discussion:

(page 13, line 16, in Discussion)

“With respect to asymmetric subgenome evolution, previous studies have reported a positive correlation of sequence similarity between ohnologs and similarity in ohnolog expression patterns¹¹. Plant genome analysis propose a “trade-off” hypothesis of asymmetric subgenome evolution in which epigenetic silencing of transposable elements regulate gene expressions^{55,56}. Despite the unknown mechanism of asymmetric subgenome evolution observed in vertebrates including African clawed frogs, it would be interesting to determine whether the observed asymmetric subgenome evolution is consistent with this hypothesis.”

Q6 cell type specific expression

6. At one point the authors state that there are 632 ohnolog pairs with cell type specific expression, but Figure 3a indicates that there are > 1000. I understand these differences are from pseudo-bulk vs cluster-based analyses, but it is quite confounding in the text. The authors should consider ways to clarify the different gene sets utilised.

As reviewer #3 points out (this comment is similar to comment 5 of reviewer #2), our description of cell type-specific ohnolog pair groups may be confusing. We chose 1070 ohnolog pairs (page 6, line 9) based on cell-type specificity of the total expression of L- and S-ohnologs to find overall cell type-specific genes; in contrast, we selected the 632 ohnolog pairs based on simple individual expression of L- or S-ohnologs to identify sub/neo-functionalized ohnolog pairs. To clarify how we selected ohnolog pairs, we added the following text to the Results:

(page 8, line 11 in the Results)

“We found that 1,070 ohnolog pairs (9.3%) showed cell type specific expression profiles (cell-type specificity of the total expression of L- and S-ohnologs)”

(page 9, line 31 in the Results)

“First, we selected the cell type specific ohnologs among the 11,444 ohnolog pairs by searching for ohnologs with significantly higher expression in a particular cell-type (cell-type specificity of individual expression of L- or S ohnolog).”

Q7 PCA

7. The authors state that they use UMAP to compare the clustering based on L- or S-ohnologs

only, however in the methods it is unclear if for these figures they also perform clustering and PCA using those same gene sets (or if they use the cluster labels from the initial clustering with all genes but with a different projection only). Sharing of scripts used for analysis would alleviate these issues. Additionally, it appears the authors used some sort of integration method to compare with zebrafish, but there are no details of this in the methods.

We performed PCA using those same gene sets. We added this result to the revised manuscript (Figure S3). In addition, we shared the scripts used for analysis on GitHub. We added the text to the revised manuscript as follows:

(page 19, line 10 in the Data Availability Section)

“The scripts used for analyses in the current study are available at GitHub (https://github.com/ironman-tetsuo/scRNA-seq_Goldfish_retina).”

(Supplemental Figure S3a Legends)

“PCA plots of the first 10 components of scRNA-seq data from Wakin goldfish retina. The 12 retinal cell types formed distinct clusters.”

Q8 Fisher's exact test

8. The authors could perform an enrichment test for whether or not S/L-biased genes are more likely to be cell-type specific (or not) than a random set of genes (Figure 3a).

We performed an enrichment test and the following added data:

(page 8, line 18 in the Results)

“We performed an enrichment test to determine whether S/L-biased genes show more cell-type specific expression than a random set of genes. The result showed no significant differences between them (Fisher's exact test, $p = 0.093$).”

Q9 potential over-expression of the S-copy

9. Given the global bias towards expression from the L-subgenome, to me it would seem that equal expression of L- and S-orthologs would indicate a potential over-expression of the S-copy. Can the authors somehow reconcile the global biases with the biases at the gene level?

Thank you for pointing this out. If most ohnolog pairs showed L-biased expression, we should consider a potential over-expression of the S-copy, as reviewer #2 mentioned. However, our new data suggest that most individual ohnolog pairs show no biased expression ($L = S$). We plotted the L/S ratio for each ohnolog pair in each cell-type (Supplemental Figure S4c), and observed that the L/S peak is centered at zero, suggesting that most ohnolog pairs show no biased expression. There is no extreme distortion that the center of the L/S is shifted. This contrasts with the right-shifted peak observed in Fig. 2c (L/S of total gene expression). This result suggests that ohnolog pairs show globally L-biased in terms of total gene expression, but not at most individual gene levels. We added these results to the revised manuscript as follows:

(page 7, line 33, in the Results)

“We plotted the L/S expression ratio for each ohnolog pair in each cell-type (Fig. S4c). The L/S peak is centered at zero, suggesting that most ohnolog pairs show no biased expression. This contrasts with the right-shifted peak observed in Fig. 2c (L/S of total read counts). This result suggests that ohnolog pairs show globally L-biased in terms of total gene expression, but not at most individual gene levels.”

(Supplemental Figure S4c Legends)

“Histogram of \log_2 transformed L/S ratios. The L/S peak is centered at zero, suggesting that the most ohnolog pairs show no biased expression. This contrasts with the right-shifted peak observed in Fig. 2c (L/S of total gene expression).”

REVIEWERS' COMMENTS:

Reviewer #1 (Remarks to the Author):

The authors have responded sincerely to my comments. In addition, they have responded in detail to the other two reviewers' comments on the statistical analysis. Considering that this type of single-cell analysis for goldfish has not been published before, I believe this will contribute to the future development of the field of the evolutionary studies of genome and gene expressions. Therefore, we hope that this paper will be published soon.

Reviewer #2 (Remarks to the Author):

In the revised version, the authors have addressed most of the concerns I raised. It would have been great to see deeper analysis of the ATAC-seq data, but the manuscript is complex as it is.

Reviewer #3 (Remarks to the Author):

The authors have addressed my comments and the comments of the other reviewers. They have provided the additional information on their methodology as requested, some additional analyses that improves their results, and extensively changed the language and interpretation of their findings where appropriate.

I endorse the publication of this study.